# S-Prompts Learning with Pre-trained Transformers: An Occam's Razor for Domain Incremental Learning

**Yabin Wang**[1,2][*] **Zhiwu Huang**[2][†] **Xiaopeng Hong**[3, 4, 1][‡]
[1]Xi'an Jiaotong University, P. R. China
[2]Singapore Management University, Singapore
[3]Harbin Institute of Technology, P. R. China
[4]Pengcheng Laboratory, P. R. China
`iamwangyabin@stu.xjtu.edu.cn, zzhiwu.huang@gmail.com, hongxiaopeng@ieee.org`

## Abstract

State-of-the-art deep neural networks are still struggling to address the catastrophic forgetting problem in continual learning. In this paper, we propose one simple paradigm (named as S-Prompting) and two concrete approaches to highly reduce the forgetting degree in one of the most typical continual learning scenarios, i.e., domain increment learning (DIL). The key idea of the paradigm is to learn prompts independently across domains with pre-trained transformers, avoiding the use of exemplars that commonly appear in conventional methods. This results in a win-win game where the prompting can achieve the best for each domain. The independent prompting across domains only requests one single cross-entropy loss for training and one simple K-NN operation as a domain identifier for inference. The learning paradigm derives an image prompt learning approach and a novel language-image prompt learning approach. Owning an excellent scalability (0.03% parameter increase per domain), the best of our approaches achieves a remarkable relative improvement (an average of about 30%) over the best of the state-of-the-art exemplar-free methods for three standard DIL tasks, and even surpasses the best of them relatively by about 6% in average when they use exemplars. *Source code is available at* `https://github.com/iamwangyabin/S-Prompts`.

## 1   Introduction

State-of-the-art deep neural networks are capable of performing impressively on a wide variety of individual machine learning tasks. However, learning multiple tasks in sequence, generally called *continual learning*, remains a substantial challenge for deep learning. When trained on a new task, standard neural networks typically forget most of the knowledge related to previously learned tasks. This is a well-known phenomenon named as *catastrophic forgetting*.

There are three common continual learning scenarios. Task-incremental learning (TIL) is provided with task indexes for inference and thus using task-specific neural networks could overcome forgetting. Class-incremental learning (CIL) increases classes in sequence with task indexes being unknown for inference. Nevertheless, the classes are generally from the same domain, which more or less decreases the challenge. In this paper, we focus on exploring domain-incremental learning (DIL), where classes keep the same but the involved domains commonly vary a lot in sequence, with task indexes being not provided for inference. The larger domain gaps result in more severe forgetting.

---

[*]This work is done when Yabin Wang visits Singapore Management University
[†]Corresponding Author
[‡]Corresponding Author

36th Conference on Neural Information Processing Systems (NeurIPS 2022).

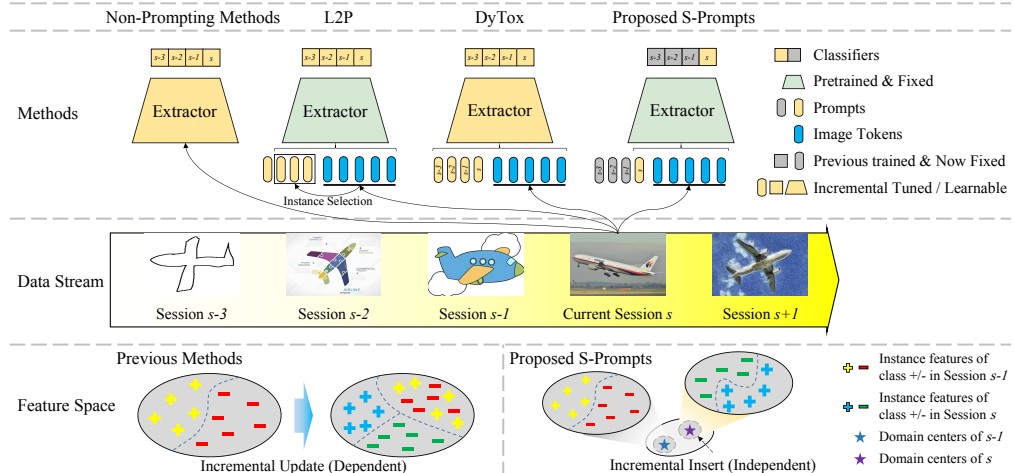

Figure 1: Comparison of the proposed S-Prompts paradigm against non-prompting methods, and prompting methods (L2P [55], DyTox [14]). Exiting methods generally learn sequential tasks/sessions/domains dependently, producing a single feature space where the classes from different domains are less separable (more forgetting or worse transferring). In contrast, the proposed S-Prompts learn the tasks independently. This paradigm leads to one subspace for every domain, making the classes more separable (less/no forgetting and better transferring).

To alleviate catastrophic forgetting, numerous methods have been proposed for continual learning. One of the most effective methods is to leverage a buffer of exemplars from old tasks to perform either a rehearsal or a distillation when tuning the whole network to learn new tasks (see Non-Prompting Methods in Fig.1). Both the rehearsal and the distillation typically decrease the forgetting degree. However, it is often more desired that the exemplars of old tasks are not stored for better data security and privacy. Moreover, storing a large amount of exemplars might run out available memories. Therefore, our focus is shifted to the challenging *exemplar-free DIL* in this paper.

One of the most promising solutions to the exemplar-free DIL problem is to learn a set of *prompts*[4] over transformers that achieve the state-of-the-arts in a wide range of areas. In this solution, the domain-specific knowledge is stored by a prompt pool, and hence a rehearsal buffer is no longer mandatory to mitigate forgetting. For instance, two recent prompting methods [14, 55][5] aim at *learning task/domain-specific prompts dependently across domains* in an incremental update fashion. In [14], the prompts are added task by task, and the newly added prompts share the same task-attention learning with the old prompts. This requests a joint tuning on all the prompts and the entire transformer using a distillation loss to keep a balance between the new prompting and the old ones. In [55], a pool of prompts is initialized at the right beginning, and a set of the prompts from the pool are selected for each instance. The instance-level prompts generally share task-level knowledge among any similar instances, by jointly tuning the selected prompts and the whole classifier, with the pre-trained transformer being fixed. While using different prompting approaches, [14, 55] both follow a commonly-respected principle that the prompting should keep sharing across tasks/domains. However, the sharing-driven dependent prompt tuning paradigm as well as many of the non-prompting paradigms generally result in *a tug-of-war (or a zero-sum game)*, in which one side's gain is equivalent to the other's loss, as studied in [47, 30]. In other words, these paradigms generally accumulate new knowledge in the same feature space, which is very likely to mix up the subspaces of old/new knowledge resulting in less separable classes (more forgetting or worse transferring) (Fig.1).

In this paper, we explore a rule-breaking idea to instead play *a win-win game, i.e., learning the prompts independently across domains so that the prompting can achieve the best for each domain.* The excellent generalization capability of the transformers and the strong transfer learning ability of the recently appeared prompting mechanisms allow for the realization of this idea. In particular, we introduce a new learning paradigm that learns the prompts independently domain by domain, and incrementally inserts the learned prompts into a pool. As shown in Fig.1, compared to the other prompting and non-prompting methods, the new paradigm suggests merely tuning the current domain-

---

[4]For simplicity and consistency, we use 'prompt' to represent 'prompt token' for context/domain knowledge encoding, and we utilize 'token' to denote those normal tokens on images and classes, throughout the paper.

[5]A concurrent work [12] learns class-level prompts dependently on transformers for the CIL problem.

related prompts and classifiers with all the rest unrelated network components being fixed. The frozen pre-trained transformer is capable of extracting general features, and the independent prompting reaches the best fit on each domain. The independent prompting is able to generate a feature space where each subspace is spanned by one single domain, and all the subspaces are less overlapped (see Feature Space in Fig.1). This could lead to no forgetting across domains. The learning process only requests the most naive cross-entropy loss for supervision. The inference is also simple and efficient. It simply uses K-NN to search for the nearest domain centers generated by K-Means on the training data to the features of the given test samples, followed by prepending the learned domain-associated prompts with the image tokens to the transformer for the final classification. As $S$ domain-related prompts will be finally learned independently, where $S$ is the total number of domains, we name the proposed paradigm as *S-Prompts* or *S-Prompting* for simplicity throughout the paper. We hope the proposed S-Prompting becomes an *Occam's Razor* (i.e., a simple and elegant principle) for DIL.

The S-Prompting paradigm can be applied to any pre-trained transformers. In this paper, we exploit two approaches of *S-Prompts* learning. One is based on vision transformer (ViT) [13], and the other is based on Contrastive Language-Image Pre-Training transformer (CLIP) [45]. Particularly, we suggest the CLIP-based approach that exploits a new language-image prompting scheme. The key idea is to prompt the pair of language-image transformers synchronously with a pair of learnable language-image prompts at the two ends. This highly enhances CLIP's transfer learning ability on a variety of new domains. As a result, this approach owns a great ability to reach a favorable feature space where the subspaces of different domains are pushed far away from each other and the classes are separated clearly, leading to highly impressive DIL performances. To the best of our knowledge, this is the first time to introduce the *paired language-image prompting scheme* in the literature.

The contributions of this paper can be summarized as follows:

- We introduce a simple and effective S-Prompts learning paradigm with pre-trained transformers for the DIL problem. This paradigm breaks the commonly-respected principle in continual learning.
- We exploit an Image S-Prompts (S-iPrompts) learning approach and a novel Language-Image S-Prompts (S-liPrompts) learning approach, based on two different pre-trained transformers.
- The best S-Prompts surpasses the best of the state-of-the-art exemplar-free methods significantly (an average of *30%* relative improvement) for three standard DIL benchmark datasets, and even outperforms them relatively by an average of *6%* when they use exemplars. S-Prompts can scale to a large number of domains with a tiny parameter increase, e.g., *0.03%* per domain in S-liPrompts.

## 2   Related Work

**Continual Learning.** Numerous methods have been exploited to address catastrophic forgetting. For instance, memory-based methods alleviate forgetting by either replaying a small set of examples from past tasks saved in a memory [48, 46, 39, 7, 21, 25, 9] or replaying synthesized data from previous tasks using generative models [50, 53]. Distillation-based methods (e.g., [35, 46, 20, 5, 56, 58, 52, 37, 41, 11]) apply the technique of knowledge distillation [19] to mitigate catastrophic forgetting. In addition, there exist other kinds of methods like gradient-based methods (e.g., [47, 60, 16, 49, 51, 54]), regularization-based ones (e.g. [29, 61, 2, 1]), and expansion-based ones (e.g. [57, 31, 23]).

Recently, some of work like [40, 28, 26, 33, 55, 17, 64] adapts the CIL methods to the DIL problems such as continual deepfake detection, where each domain distinguishes between deepfakes and reals. Following them, one of our approaches uses the common classifiers from the CIL methods. While each domain share the same classes, the incremental classifiers simply treat them as difference classes. The finer-grained task could make networks more powerful [40].

**Prompt Learning.** The general idea of prompting is to learn a function to modify the input texts or images, such that the language or image model obtains additional information about the task [55]. There have been emerging a bunch of prompting methods, such as [32, 34, 24, 62, 63, 3, 22, 55, 14], to name a few[6]. Nevertheless, these methods merely learn either image-end prompts or language-end prompts (at most using conditions from image-end), while ours explicitly learns joint language-image prompting. Recently, L2P [55] and DyTox [14] exploit dependent prompting methods for continual learning, which leads to a tug-of-war issue, as discussed in Sec.1. Our suggested S-Prompting paradigm instead learns task/domain associated prompts independently for a win-win gain.

---

[6]For a comprehensive survey on the prompting methods, please refer to [36].

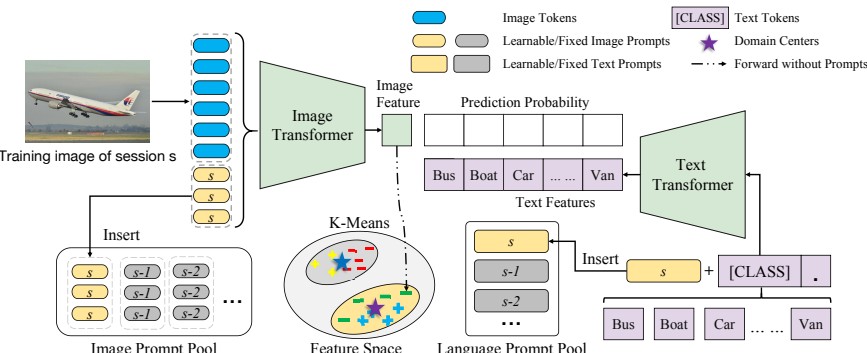

Figure 2: Illustration of the proposed language-image S-Prompts (S-liPrompts) learning approach. $s, s-1, s-2$ denote the session/domain indexes. For the inference, we simply adopt the following steps: 1) feeding the tokens of a given test image into the image transformer to get the image feature, 2) using K-NN to search for the nearest centroid obtained by K-Means on training data to the feature of the given test image, 3) feeding the image token and the nearest centroid (domain center) related image prompt into the image transformer, with the centroid related language prompt and class tokens being fed into the text transformer, 4) computing the language-image prediction probabilities for the final classification. Note that both K-Means and K-NN are performed on the image feature space of the fixed pre-trained image transformer without using any prompts.

## 3 Proposed Approach

### 3.1 Problem Formulation and S-Prompts Learning

We focus on the problem of exemplar-free domain incremental learning (DIL). In a common and practical DIL scenario, the model is expected to learn knowledge of various domains in sequence to finally become a universal expert for all the domains. Formally, at domain $s$, the model accesses to the incoming data $\mathbb{D}_s = \{x_i^s, y_i^s\}_{i=1}^{N_s}$ where $x_i^s$ is the $i$-th input image from domain $s$ and $y_i^s$ is the corresponding label, and the total number of samples at domain $s$ is $N_s$. In the entire incremental learning stream, $S$ domains are presented sequentially, and each domain $s$ has data $\mathbb{D}_s$ from different or even highly heterogeneous domains (Fig.1). The set of class categories the model observed is $\mathbb{C}$, and in DIL all categories are identified from the beginning and will not increase along the incremental learning stream. The exemplar-free DIL requests zero exemplar from the learned sessions during the training and the inference for better data security, privacy and cheaper memory consumption.

To address the exemplar-free DIL problem, we propose an S-Prompts learning paradigm that is simple and effective. The key idea is to learn prompts domain by domain independently with pre-trained transformers. The pre-trained transformers are fixed to extract highly expressive low-level features, and the prompts are tuned to encode the high-level domain knowledge. In this case, each domain knowledge will be encoded by one prompt or one set of prompts. This not only avoids the use of exemplars but also highly reduces the catastrophic forgetting. Accordingly, our focus is on learning the prompts of $S$ domains or so-called *S-Prompts*. As shown in Fig.1, the suggested S-Prompting merely tunes the current domain related prompts and classifier with all the rest neural network components, including the transformer, the current session unrelated prompts and classifiers, being fixed. In this case, each set of domain-specific prompts are learned independently from the other domains related prompts. This requests a domain identifier for inference. In particular, we apply K-Means to store the centriods for each domain during training, and K-NN to search for the nearest centroid of the given test image feature to identify its corresponding domain during inference. Note that both K-Means and K-NN are performed on the feature space of the fixed pre-trained transformer without using any prompts. Finally, we prepend the domain associated prompts and the image tokens to the transformer for image classification. The inference steps are detailed in the caption of Fig.2. Since DIL's domain shifts are generally explicit and the state-of-the-art transformers are commonly good at image classification, the suggested domain identifier performs well in standard DIL tasks. Due to the strong generalization capability of transformers, the suggested S-Prompting still performs well for those cases with incorrect domain identification or those on unseen domains. Thanks to this mechanism, the suggested S-Prompting can achieve the best transferring for each domain, as well as the least forgetting. Moreover, the proposed S-Prompting only requests the most naive cross-entropy loss for the guidance on classification, which further enhances the paradigm's simplicity.

In this paper, we mainly study two concrete S-Prompting approaches: 1) *Image S-Prompts (S-iPrompts) learning* based on the pre-trained vision transformer (ViT) [13], and 2) *Language-Image S-Prompts (S-liPrompts)* based on the pre-trained Contrastive Language-Image Pre-Training transformer (CLIP) [45]. In essential, the main differences between these two approaches are two-folds: 1) prompt design, and 2) classifier design. We detail them below for these two approaches.

## 3.2 Image S-Prompts (S-iPrompts) Learning Approach

**Prompt Design.** In the approach of S-iPrompts, for one domain $s$, we suggest using an independent set of continuous learnable parameters (i.e., one image prompt) $P_s^i \in \mathbb{R}^{L_i \times D_i}$ as a part of inputs to the pre-trained ViT, where $L_i \in \mathbb{R}$ and $D_i \in \mathbb{R}$ indicate the prompt's length and embedding dimension respectively. As shown in Fig. 2, the whole embedding of any single image from domain $s$ follows the format of $x = [x_{img}, P_s^i, x_{cls}]$, where $x_{img}$ denotes the image tokens and $x_{cls}$ represents the pre-trained class tokens of ViT. The extended image embedding is used as an input of the transformer blocks as normal ViT does. When trained on a new domain $s + 1$, a new independent set of images prompts $P_{s+1}^i$ is added. Hence, learning all the domains sequentially results in a pool of domain-wise prompt pool. The image prompt pool can be defined as $\mathcal{P}^i = \left\{ P_1^i, P_2^i, ..., P_S^i \right\}$, where $P_s^i \in \mathbb{R}^{L_i \times D_i}$ is a single set of prompts belonging to domain $s$, and $S$ is the total number of domains.

**Classifier Design.** The classifier of S-iPrompts is a normal fully connected (FC) layer, which is made of a linear projection parameterized by $[W_s, b_s]$, where $W_s \in \mathbb{R}^{C \times D_i}$, $b_s \in \mathbb{R}^C$, $D_i, C$ indicate the embedding dimension and the total class number respectively, and $s$ is the domain index. Formally, the classifier output is calculated by

$$\hat{y} = \sigma(W_s f(x) + b_s), \tag{1}$$

where $f(x)$ represents the output image feature of the ViT encoder $f$, and $\sigma$ is the softmax activation operation, $\sigma(z_j) = \frac{e^{z_j}}{\sum_{k=1}^{C} e^{z_k}}$, for $j = 1, 2, \ldots, C$ ($C$ indicates the number of classes). Every learning session owns such one classifier, and thus we use a classifier pool $\mathcal{P}^{fc} = \{[W_1, b_1], [W_2, b_2], ..., [W_S, b_S]\}$ to store all these classifiers that are independent to each other. During the inference, once the domain center is selected, the associated prompts and the corresponding classifier will be chosen for the prediction on each given test image.

## 3.3 Language-Image S-Prompts (S-liPrompts) Learning Approach

The S-liPrompts approach aims at exploiting an effective prompting scheme on CLIP [45] so as to transfer the pre-trained CLIP to a variety of domains. As illustrated in Fig.2, the CLIP model consists of two transformers for image and text encoding respectively. The CLIP model was trained on an enormous amount of image-text pairs in a contrastive learning manner, which aligns the pair of language-image transformers very tightly. Hence, it is essential to prompt the pair of CLIP transformers synchronously for an effective transfer learning. To this end, we follow the paradigm of S-Prompts to exploit a joint language-image prompting scheme. The scheme learns a pair of an image-end prompt and a language-end prompt domain by domain, with the pre-trained CLIP always being fixed. On one hand, fixing CLIP enables the prompting model to inherit the extraordinary capability of CLIP in terms of zero-shot learning and few-shot learning [62]. On the other hand, the joint prompting can make the image-end transformer has an excellent fit into any upcoming domains that might be highly different from the pre-training data. This is mainly because synchronously prompting the language-end transformer could get more semantic information and more domain-aware (Fig.4).

**Prompt Design.** The design of image-end prompts for S-liPrompts is the same as that of S-iPrompts. For the design of language-end prompts, we suggest using a learnable context prompt to replace the artificially designed prompts used by the original CLIP. This design can better adapt pre-trained vision-language models to downstream tasks, especially for few-shot learning. In particular, for the learning domain $s$, we use $M$ learnable vectors $v$ as the context prompt $P_s^l = \{v_1, v_2, ..., v_M\} \in \mathbb{R}^{L_l \times D_l}$, where $L_l, D_l$ are the prompt's length and embedding dimension respectively. We regard it as text prompts corresponding to image prompts used in ViT. For the $j$-th class, the whole input $t_j$ of the text encoder is of the format $t_j = \{P_s^l, c_j\}$, where $c_j$ is the word embeddings of the $j$-th class names [CLASS] and has the same dimension of $D_l$. The text prompts are also associated with one single domain, and they are independent to those from the other domains. After learning all the sequential domains, a pool of text prompts is formed as $\mathcal{P}^l = \{P_1^l, P_2^l, ..., P_S^l\}$, where $S$ is the domain number.

The learnable context prompt can better adopt the pre-trained CLIP to various domains, which is an essential property for incremental learning.

**Classifier Design.** The text encoder $g$ of CLIP receives the input $t_j$ defined above, and produces a vectorized representation $g(t_j)$, where $j = 1, 2, \ldots, C$, and $C$ denotes the class number. The contrastive language-image pre-training of CLIP uses a contrastive loss to learn a joint embedding space for vision and language models. One of its objectives is to maximize the cosine similarity of each image with its matched text and minimize the cosine similarities with all other unmatched texts. The other objective is computed in a similar fashion for each text. Let $f(x)$ be the image feature generated by the image encoder $f$, and $\{g(t_j)\}_j^C$ is a set of weight vectors produced by the text encoder $g(.)$, each $g(t_j)$ representing the text feature of $j$-class. The prediction probability is computed as

$$p(y_j|x) = \frac{\exp(\langle f(x), g(t_j)\rangle)}{\sum_{k=1}^{C} \exp(\langle f(x), g(t_k)\rangle)}, \tag{2}$$

where $\langle ., .\rangle$ is the cosine similarity, and $\langle x, y\rangle = \frac{x \cdot y}{|x||y|}$. We conduct the softmax activation operation $\sigma$ over the result of Eq.2 to obtain the final logits for image classification.

Like S-iPrompts, S-liPrompts also grows the pool of domain-based classifiers (Fig.2). Compared with the FC-based classifiers in S-iPrompts, there are two main advantages using the CLIP-based classifier in S-liPrompts. Firstly, the language model provides more domain information using the domain-specific text prompts for DIL, because it generally leads to more domain-separable features (see Fig.4). Secondly, in the incremental process, only one domain context needs to be stored for each session, while the class names are shared, which highly reduces the memory consumption.

## 4 Experiments

**Datasets.** We perform experiments on three standard DIL benchmark datasets. They are CDDB [33], CORe50 [38], and DomainNet [43]. CDDB [33] is a dataset for continual deepfake detection, which is a challenging DIL task. It designs Easy, Long, and Hard tracks. We select the most challenging track (i.e., the Hard track). The track requests learning on 5 sequential deepfake detection domains (about 27,000 images), which are GauGAN, BigGAN, WildDeepfake, WhichFaceReal, and SAN respectively. CORe50 [38] is a widely used dataset for continual object recognition. It has 50 categories from 11 distinct domains. The continual learning setting uses 8 domains (120,000 images) for incremental training and the data from the rest (i.e., unseen) domains as test set. DomainNet [43] is a dataset for domain adaptation and DIL, which has 345 categories and roughly 600,000 images. Images from DomainNet is split into 6 domains. The DIL setup on DomainNet is the same as that of CaSSLe [17]. We follow [33] to report the average forward detection accuracy and the average forgetting degree on CDDB-Hard. For CORe50 and DomainNet, we follow [55] and [17] to present the average forward classification accuracy.

**Methods.** We compare the proposed S-Prompts methods (S-iPrompts, S-liPrompts) against the state-of-the-art CIL/DIL methods. They include non-prompting methods EWC [29], LwF [35] ER [8], GDumb [44], BiC [56], DER++ [4] and Co2L [6], as well as the prompting methods DyTox [14] and L2P [55] and self-supervised learning method CaSSLe [17]. Among them, since EWC [29], LwF [35], L2P [55], CaSSLe [17] can work for examplar-free DIL, we regard them as our real competitors. For DyTox and L2P, we use their official implementations by tuning their parameters for better performances. To compare fairly, we use the same ViT model (i.e., ViT-B/16 [13]) for most of the real competitors [7] as well as our S-iPrompts/S-liPrompts. The proposed S-liPrompts's text-end encoder is the same as that of CLIP [45]. For more implementations details and studies, please refer to Appendix.

**Overheads.** For each domain, our proposed S-Prompting methods additionally use a set of prompts as well as a set of centroids from K-Means. For S-liPrompts, the overhead in memory is merely of $+0.03\%$ per domain. Compared to the other prompting methods DyTox and L2P, S-liPrompts enjoys much less memory overhead of the growing pool of domain-based classifiers due to its CLIP based classifier design (i.e., extremely cheap prompts and shared class names) when the class number

---

[7]Among the competitors, only DyTox [14] uses a more advanced ViT model(ConViT [15]), since it fails to get promising results (e.g., the result is even inferior than that of the random guess on CDDB) using ViT-B/16 [13].

Table 1: Results on CDDB-Hard for deepfake DIL. **Bold**: best exemplar-free DIL results, Underline: second best exemplar-free DIL results. Upper-bound: supervised finetuning on the i.i.d. data of all tasks, which is usually regarded as the upper bound performance a method can achieve [55]. * denotes results copied from [33].

| Method | Buffer size | Average Acc (↑) | Forgetting (↑) |
|---|---|---|---|
| LRCIL [42] | | 76.39* | -4.39* |
| iCaRL [40] | 100/class | 79.76* | -8.73* |
| LUCIR [20] | | 82.53* | -5.34* |
| LRCIL [42] | | 74.01* | -8.62* |
| iCaRL [40] | | 73.98* | -14.50* |
| LUCIR [20] | 50/class | 80.77* | -7.85* |
| DyTox [14] | | 86.21 | -1.55 |
| EWC [29] | | 50.59 | -42.62 |
| LwF [35] | | 60.94 | -13.53 |
| DyTox [14] | | 51.27 | -45.85 |
| L2P [55] | 0/class | 61.28 | -9.23 |
| S-iPrompts (ours) | | 74.51 | -1.30 |
| S-liPrompts (ours) | | **88.65** | **-0.69** |
| Upper-bound (S-iPrompts) | - | 85.50 | - |
| Upper-bound (S-liPrompts) | - | 91.91 | - |

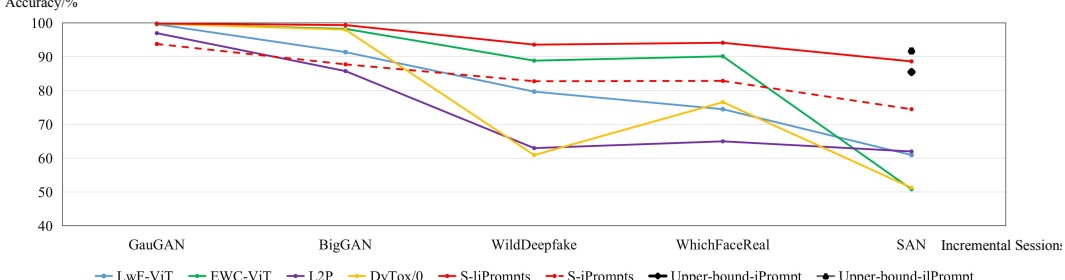

Figure 3: Accuracy curves of the competing methods and the proposed S-Prompts on CDDB-Hard.

is large (e.g., 345 classes in DomainNet). Accordingly, the proposed methods own an excellent scalability. On the other hand, S-liPrompts takes 11.84 millisecond (ms) in average to classify an image on a single RTX-3090, while the pre-trained CLIP model takes an average of 9.37ms. This shows that the additional domain identification merely takes around 2.5ms, which is affordable for the real-world scenarios. The appendix presents more details about the overheads.

**Results.** The results in Table 1, Table 2 and Table 3 demonstrate that the proposed S-iPrompts and S-liPrompts significantly outperform the other exemplar-free methods including the two recent prompting methods L2P and DyTox. We can compute that the proposed S-liPrompts obtains a considerable relative improvement (an average of roughly 30%) over the best of these methods in terms of forward classification accuracy. From Table 1 and Fig.3, we could also find the proposed S-Prompting methods' forgetting degrees are much less than those of the others. On the other hand, compared to those methods using exemplars of old tasks, our S-liPrompting still enjoys the clear superiority (an average of about 6% relative improvement) over the best of them. Besides, it is fairly surprising that our S-liPrompts even surpasses its upper-bound (joint training version) on DomainNet that contains highly heterogeneous domains. This is very likely that the joint learning cannot handle the large domain shifts well, and our independent prompting avoids this issue. Another interesting phenomena is that our S-Prompts still enjoy the extraordinary generalization capability (both S-iPrompts and S-liPrompts clearly surpass others) on the unseen domains of CORe50 where all the tested domains do not appear in the incremental training. The inferior performance of L2P on DomianNet might be due to the less transfer learning capability of its dependent instance-level prompting (prompt sharing) on heterogeneous domains. While DyTox reaches closer to the proposed S-liPrompts on DomainNet, it requests a large memory overhead (i.e., 50×345=17,250, where 50, 345 are the exemplar number per/class and the class number respectively) for the storage of exemplars[8].

**Ablations.** The studies on the cases of fixing either FC length/weights or Prompt length/weights in Table 4 justifies the necessity of using the proposed growing classifier pool and independent

---

[8]DyTox [14] often fails without using exemplars, as it requests a distillation on exemplars for better prompting.

Table 2: Results on CORe50 for object DIL. **Bold**: best exemplar-free DIL results, Underline: second best exemplar-free DIL results. Upper-bound: supervised finetuning on the i.i.d. data of all tasks, which is usually regarded as the upper bound performance a method can achieve [55]. * indicates results copied from [55].

| Method | Buffer size | Average Acc (↑) |
|---|---|---|
| ER [8] | | $80.10_{\pm 0.56*}$ |
| GDumb [44] | | $74.92_{\pm 0.25*}$ |
| BiC [56] | | $79.28_{\pm 0.30*}$ |
| DER++ [4] | 50/class | $79.70_{\pm 0.44*}$ |
| $Co^2L$ [6] | | $79.75_{\pm 0.84*}$ |
| DyTox [14] | | $79.21_{\pm 0.10}$ |
| L2P [55] | | $81.07_{\pm 0.13*}$ |
| EWC [29] | | $74.82_{\pm 0.60*}$ |
| LwF [35] | | $75.45_{\pm 0.40*}$ |
| L2P [55] | 0/class | $78.33_{\pm 0.06*}$ |
| S-iPrompts (ours) | | $83.13_{\pm 0.51}$ |
| S-liPrompts (ours) | | **$89.06_{\pm 0.86}$** |
| Upper-bound (S-iPrompts) | - | $84.01_{\pm 0.53}$ |
| Upper-bound (S-liPrompts) | - | $93.19_{\pm 0.21}$ |

Table 3: Results (task-agnostic test results) on DomainNet for object DIL. **Bold**: best exemplar-free DIL results, Underline: second best exemplar-free DIL results. $\dagger$: the total buffer size is $50 \times 345 = 17,250$, where 345 is the total number of DomainNet classes. It is a large memory overhead. Upper-bound: supervised finetuning on the i.i.d. data of all tasks, which is usually regarded as the upper bound performance a method can achieve [55]. * indicates results copied from [17].

| Method | Buffer size | Average Acc (↑) |
|---|---|---|
| DyTox [14] | 50/class$^\dagger$ | 62.94 |
| EWC [29] | | 47.62 |
| LwF [35] | | 49.19 |
| SimCLR [10]-CaSSLe [17] | | 44.2* |
| BYOL [18]-CaSSLe [17] | | 49.7* |
| Barlow Twins[59]-CaSSLe [17] | | 48.9* |
| Supervised Contrastive [27]-CaSSLe [17] | 0/class | 50.9* |
| L2P [55] | | 40.15 |
| S-iPrompts (ours) | | 50.62 |
| S-liPrompts (ours) | | **67.78** |
| Upper-bound (S-iPrompts) | - | 63.22 |
| Upper-bound (S-liPrompts) | - | 64.08 |

image/language prompting for S-iPrompts/S-liPrompts. We also study the case that the domains are randomly selected for the inference. It is surprising that the accuracy (80.12%) does not drop a lot and is still better than those of the state-of-the-art methods. In fact, the domain identification accuracies are not high using either ViT-based or CLIP-based features, as shown in Table 5. Both of these two results show that the proposed S-Promptings are not fundamentally influenced by the domain identification accuracy. As shown in Fig.4, the intrinsic reason might be that the proposed S-iPrompts and S-liPrompts actually result in a clearer domain separation compared to the pre-trained ViT and CLIP, and also they inherit the excellent generalization capabilities of the used transformers. This is justified by the result of S-liPrompts ($K = 5$ & Zero-shot) that is merely trained on the first domain and tested on all the domains. Through the study on different $K$ values in K-Means, we find that the proposed S-liPrompts works very stably for different $K$ values. Hence, we consistently use $K = 5$ for all the experiments on CDDB-Hard, CORe50 and DomainNet.

Apart from the number of centers $K$ for K-Means (studied in the Table 4), there are three other key hyper-parameters for the proposed S-Prompts, including the length of an image prompt $L_i$, the length of a language prompt $L_l$ (only for S-liPrompts), and the value of $K$ in K-NN selection algorithm. We perform further ablations on the proposed S-liPrompts by keeping one of these hyper-parameters varied and fixing the rest hyper-parameters on CDDB-Hard. Fig.5 (Top Left) illustrates the accuracy curve of using different image prompt lengths. We can see that too small $L_i$ has negative impact, which means that the prompts have not much ability to transfer the pre-trained model to downstream tasks. However, too large $L_i$ can not help the model to get better performance. Fig.5 (Top Right) illustrates the accuracy curve of using different language prompt length. Compared with the study

Table 4: Results of ablating components of the proposed two S-Prompts (i.e., S-iPrompts, S-liPrompts) methods on CDDB-Hard for exemplar-free deepfake DIL. Different $K$ values for K-Means.

| Method | Average Acc ($\uparrow$) | Forgetting ($\uparrow$) |
|---|---|---|
| S-iPrompts (fixed FC length) | 52.07 | -28.89 |
| S-iPrompts (fixed FC weights) | 72.13 | -0.86 |
| S-iPrompts ($K = 5$ & Zero-shot) | 62.08 | - |
| S-iPrompts (final, $K = 5$) | 74.51 | -1.30 |
| S-liPrompts (fixed Language & Image Prompt length) | 64.90 | -12.7 |
| S-liPrompts (fixed Language Prompt length) | 52.80 | -29.13 |
| S-liPrompts (fixed Language Prompt weights) | 72.29 | -0.99 |
| S-liPrompts ($K = 5$ & Zero-shot) | 66.57 | - |
| S-liPrompts ($K = 5$ & Domain Random Selection) | 80.12 | -1.04 |
| S-liPrompts ($K = 1$) | 88.72 | -1.09 |
| S-liPrompts ($K = 3$) | 88.55 | -0.77 |
| S-liPrompts ($K = 7$) | 88.40 | -0.82 |
| S-liPrompts ($K = 9$) | 88.32 | -0.88 |
| S-liPrompts (final, $K = 5$) | **88.65** | **-0.69** |

Figure 4: t-SNE visualization on the resulting feature spaces of pre-trained ViT, proposed S-iPrompts, pre-trained CLIP and proposed S-liPrompts on CDDB-Hard.

Table 5: Domain identification accuracy (task-wise) when $K = 5$ for K-Means and $K = 1$ for K-NN on CDDB-Hard.

| | GauGAN | BigGAN | WildDeepfake | WhichFaceReal | SAN | Average Acc ($\uparrow$) |
|---|---|---|---|---|---|---|
| ViT | 0.82 | 0.78 | 0.98 | 0.98 | 0.56 | 0.82 |
| CLIP | 0.74 | 0.68 | 0.97 | 0.97 | 0.66 | 0.80 |

Figure 5: Domain incremental learning accuracies of the proposed S-liPrompts (Top Left) using different image prompt lengths with the language prompt length being fixed as 16, (Top Right) using different language prompt lengths with the image prompt length being fixed as 10, (Bottom Left) using different $K$ values for K-NN based domain identification, and (Bottom Right) domain identification accuracies (task-agnostic) of the cases with different $K$ values for K-NN, on the CDDB-Hard dataset.

on image prompts, language prompts also favor a sufficient length (the longest one is 16 in our case). Fig.5 (Bottom Left) studies the influence of using different $K$ values in K-NN on the domain incremental learning accuracy. From the results, we can find that 1-NN works clearly better than the other cases. The main reason is that the other K-NN cases result in a clear performance drop for domain identification (Fig.5 (Bottom Right)). Therefore, we suggest using 1-NN that is the simplest and the most effective for the proposed S-Prompts.

**TIL vs. DIL.** Table 6 shows the results of the proposed S-liPrompts in the scenarios of task-incremental learning (TIL) and domain-incremental learning (DIL). TIL offers domain indexes for the inference, while DIL does not provide the information. We report task-wise average accuracy (AA) that computes the average on all the task-based accuracies, and also we present task-agnostic

Table 6: Accuracies (%) of the proposed S-liPrompts in the setups of task-incremental learning (TIL) and domain-incremental learning (DIL) on CDDB-Hard. TIL provides domain indexes for the inference phase, while DIL does not offer domain indexes for the test. AA: average accuracy. Task-wise AA: average on all the task-based accuracies. Task-agnostic AA: average on all the test images without considering the task indexes.

|     | GauGAN | BigGAN | WildDeepfake | WhichFaceReal | SAN | Task-wise AA | Task-agnostic AA |
|-----|--------|--------|--------------|---------------|-----|--------------|------------------|
| TIL | 99.85  | 99.50  | 82.21        | 96.50         | 85.56 | **92.72**  | **92.51**        |
| DIL | 99.30  | 96.75  | 82.06        | 96.25         | 68.89 | 88.65      | 91.54            |

Table 7: Accuracies (%) of the application of the trained S-liPrompts on S1-S5 to out-of-domains OOD1-OOD3 that are 3 unseen domains used in CDDB. S1-S5: GauGAN, BigGAN, WildDeepfake, WhichFaceReal, SAN, which are used to train S-liPrompts on CDDB-Hard. OOD1-OOD3: FaceForensic++, Glow, StarGAN, which are not used to trained on S-liPrompts. AA: average accuracy. Task-wise AA: average on all the task-based accuracies. Task-agnostic AA: average on all the test images without considering the task indexes. [†]: CDDB-Hard runner-up method that uses 50 exemplars per class, while ours utilizes ZERO exemplar.

|            | S1    | S2    | S3    | S4    | S5    | OOD1  | OOD2  | OOD3  | Task-wise AA | Task-agnostic AA |
|------------|-------|-------|-------|-------|-------|-------|-------|-------|--------------|------------------|
| S1 (ours)  | 99.85 | 87.75 | 49.15 | 50.50 | 50.00 | 50.92 | 63.52 | 71.10 | 65.35        | 68.43            |
| S2 (ours)  | 99.90 | 98.88 | 51.24 | 52.50 | 50.00 | 74.68 | 73.42 | 80.06 | 72.59        | 76.56            |
| S3 (ours)  | 99.90 | 98.75 | 82.16 | 52.25 | 51.11 | 73.29 | 66.83 | 78.15 | 75.31        | 77.89            |
| S4 (ours)  | 99.85 | 98.50 | 82.06 | 96.25 | 52.22 | 75.60 | 64.58 | 92.60 | 82.71        | 82.78            |
| S5 (ours)  | 99.30 | 96.75 | 82.06 | 96.25 | 68.89 | 75.69 | 64.44 | 92.46 | **84.48**    | **82.63**        |
| S5 (DyTox[†][14]) | 97.90 | 96.12 | 83.23 | 94.50 | 62.22 | 81.42 | 54.83 | 55.77 | 78.25 | 69.02 |

AA that computes the average on all the test images without considering the task indexes. Therefore, the task-agnostic AA is actually more in line with the real-world settings. The task-agnostic AAs show that S-liPrompts almost *closes the gap between DIL and TIL.*

**Generalization.** We test the generalization capability of the proposed S-liPrompts on unseen domains i.e., out-of-domain data (OOD). In detail, we apply the pre-trained S-liPrompts on CDDB-Hard's 5 domains to 3 OOD data, which are FaceForensic++, Glow, StarGAN used in CDDB [33]. Table 7 reports the results of the OOD experiment. The accuracies on the three OODs increase clearly as S-liPrompts is trained on more seen data (S1-S5). We report the accuracies in both task-wise and task-agnostic cases. The results show that our S-liPrompts has a better generalization ability to these OODs than DyTox [14], which is the runner-up on CDDB-Hard.

# 5   Discussion and Conclusion

Continual learning studies the problem of learning from a data stream from changing domains/tasks. The general principle of conventional methods is to adapt to new domains/tasks by accumulating previously acquired knowledge, while protecting previous learning from being erased. Beyond this commonly-respected principle, this paper proposes a rule-breaking paradigm, which is simple but effective, for the domain incremental learning problem. In particular, the proposed paradigm performs incremental learning without using any expert knowledge from previously learned domains except for general knowledge from pre-trained transformers. Instead, it merely learns the expert knowledge independently domain by domain, through simply prompting the state-of-the-art transformers. The independent knowledge learning mechanism avoids the traditional tug-of-war among different learning sessions and instead plays a win-win game to achieve the best for each learning session. The learned expert knowledge for each domain is finally gathered in a pool for a general inference on varying domains. Our comprehensive empirical study shows that the proposed learning paradigm enjoys a groundbreaking gain (about 30% relative improvement) over the state-of-the-art competitors as well as a great scalability to a large number of incrementally appearing domains.

There remain limitations on the suggested paradigm. For example, we do not obtain appealing results when applying it directly to the class-incremental learning (CIL) tasks. This is because these tasks are more sensitive to the incorrect domain/task identification. Nevertheless, the proposed paradigm still has a high potential to inspire a wide range of research areas. For instance, the suggested prompting can be applied directly to any transfer learning problems, as it essentially learns to prompt the transformers to any upcoming new domains. In addition, it provides an inspiration to zero-shot and few-shot learning, since the proposed S-liPrompts shows superior generalization capability on unseen domains i.e., out-of-domain data (OOD).

**Acknowledgments.** This work is funded by the Singapore Ministry of Education (MOE) Academic Research Fund (AcRF) Tier 1 grant (MSS21C002). This work is also supported by the National Natural Science Foundation of China (62076195) and the Fundamental Research Funds for the Central Universities (AUGA5710011522).

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
