# S-Prompts Learning with Pre-trained Transformers: An Occam's Razor for Domain Incremental Learning (Supplementary Material)

**Yabin Wang**[1,2]*, **Zhiwu Huang**[2]†, **Xiaopeng Hong**[3, 4, 1]‡

[1]Xi'an Jiaotong University, P. R. China
[2]Singapore Management University, Singapore
[3]Harbin Institute of Technology, P. R. China
[4]Pengcheng Laboratory, P. R. China

`iamwangyabin@stu.xjtu.edu.cn, zzhiwu.huang@gmail.com, hongxiaopeng@ieee.org`

## 1 Appendix

In the appendix, we additionally present detailed algorithms of the proposed S-liPrompts, thorough experimental settings of the competing methods and the proposed ones, more ablation experiments, as well as more discussions on related works.

### 1.1 Algorithm Details

**Inference Pipeline.** Fig.1 illustrates the inference pipeline of the proposed language-image S-Prompts (S-liPrompts). For the inference phase, we suggest the following steps: 1) feeding the tokens of a given test image into the image transformer to obtain the image feature, 2) utilizing K-NN to search for the nearest centroid obtained by K-Means on training data to the given test image, 3) feeding the image token and the nearest centroid associated image prompts into the image transformer, with the centroid related language prompts and class tokens being fed into the text transformer, 4) calculating the language-image prediction probabilities for the final classification. Note that K-NN is conducted on the image feature space of the fixed pre-trained image transformer without using any prompts. **Algorithm Details.** Alg.1 and Alg.2 show the training/testing process details of the proposed S-liPrompts.

### 1.2 Experimental Details

**Implementation details.** We implement the proposed S-Prompts in PyTorch with NVIDIA RTX 3090 GPUs. We use the same image encoder architecture of ViT-B/16 [1] for S-iPrompts and S-liPrompts across all tasks/domains. The text encoder of S-liPrompts is the same as that of CLIP [23]. In our experiments, the total class number $C$ is 2 for CDDB [12], 50 for CORe50 [18], and $345$ for DomainNet [22] respectively. For all the experiments in the main paper, the length of a single image prompt $L_i$ is 10, and the length of one language prompt $L_l$ is 16. The embedding dimension $D_i$ is 768 for both of the used ViT and CLIP. The dimension of word embedding $c_i$ and the embedding dimension of text transformer $D_l$ are both 512. The implementation of K-Means and K-nearest neighbors (KNN) are both based on scikit-learn [21]. The distance measure is the L1 distance.

**Training details.** Our S-Prompts is insensitive to the setting of hyper-parameters and we do not tune much. For all benchmarks, we adopt SGD optimizer with a momentum of $0.9$, an initial learning rate

---

*This work is done when Yabin Wang visits Singapore Management University
†Corresponding Author
‡Corresponding Author

36th Conference on Neural Information Processing Systems (NeurIPS 2022).

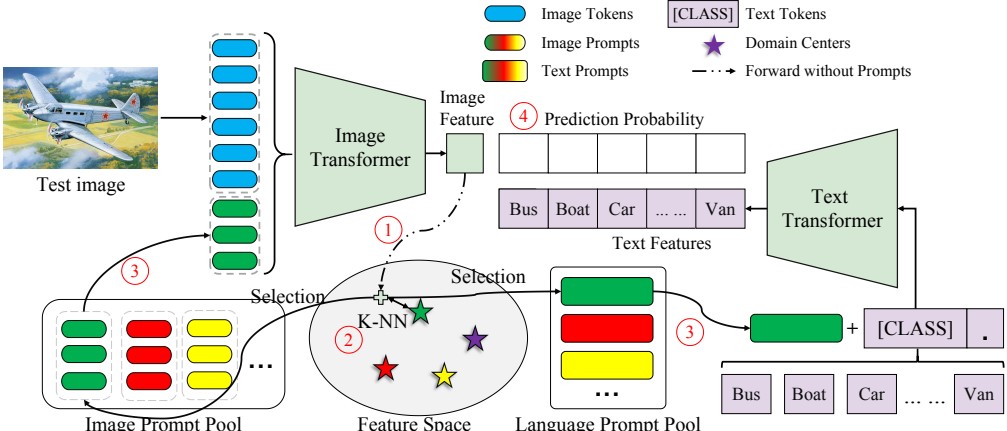

Figure 1: Illustration of the inference pipeline of the proposed S-liPrompts. The displayed indexes correspond to the following four inference steps respectively: 1) obtaining transformer feature of a given test image, 2) searching for the nearest centroid obtained by K-Means on training data to the given test image, 3) feeding the image/text tokens and the nearest centroid associated image/text prompts into the image/text transformers, 4) computing the language-image prediction probabilities for the final classification.

---

**Algorithm 1** Training process of the proposed S-liPrompts

1:  Initialization: Training data:$\mathbb{D}_s = \{x_i^s, y_i^s\}_{i=1}^{N_s}, s = 1, 2, ..., S$; Pre-trained image model: $f$; Pre-trained language model: $g$;        Class embeddings: $c_j, j$        $=$        $1, 2, ..., C$; Image Prompt Pool: $\mathcal{P}^i = \emptyset$; Language Prompt Pool: $\mathcal{P}^l = \emptyset$; Domain centroids: $\mathcal{M} = \emptyset$; Maximum Iteration: $T_{max}$; Batch size: $N_{bs}$; Number of centroids per session: $K$;

2:  **for** $s = 1, 2, ..., S$ **do**

3:      Initialize image prompt $P_s^i$ for domain $s$

4:      Initialize language prompt $P_s^l$ for domain $s$

5:      **for** $iter = 1, 2, ..., T_{max}$ **do**

6:          Extract a mini-batch samples $\{x_i^s, y_i^s\}_{i=1}^{N_b}$ from Traning Data $\mathbb{D}_s$

7:          Prepend $x_i^s$ with image prompts $P_s^i$ for image encoder by $[x_i^s, P_s^i, x_{cls}]$

8:          Prepare input with language prompts $P_s^l$ for text encoder by $t_j = \{P_s^l, c_j\}_{j=1}^C$

9:          Generate class representation by $g(t_j)$

10:          Compute the prediction probability by $p = \dfrac{\exp(\langle f([x_i^s, P_s^i, x_{cls}]), g(t_j) \rangle)}{\sum_{k=1}^C \exp(\langle f([x_i^s, P_s^i, x_{cls}]), g(t_k) \rangle)}$

11:          Calculate Cross-Entropy loss by $\mathcal{L}_B = \mathcal{L}(\sigma(p), y_i^s)$

12:          Update $\theta_{P_s^i} = \theta_{P_s^i} - \eta \triangledown_{P_s^i} \theta_{P_s^i}$

13:          Update $\theta_{P_s^l} = \theta_{P_s^l} - \eta \triangledown_{P_s^l} \theta_{P_s^l}$

14:      **end for**

15:      Initialize a temporary memory buffer $B$ to store features

16:      **for** $i = 1, 2, ..., N_s$ **do**

17:          Extract a sample $x_i^s$ from training data $\mathbb{D}_s$

18:          $B \leftarrow f([x_i^s, x_{cls}])$

19:      **end for**

20:      Run K-Means on $B$ to get $K$ domain centroids $\{m_i^s\}_{i=1}^K$

21:      $\mathcal{M} \leftarrow \{m_i^s\}_{i=1}^K$

22:      $\mathcal{P}^i \leftarrow P_s^i$

23:      $\mathcal{P}^l \leftarrow P_s^l$

24: **end for**

---

---

**Algorithm 2** Testing process of the proposed S-liPrompts

---

1: Initialization:  Test image:$x$;  Pre-trained image model:$f$;  Pre-trained language model:$g$; Class embeddings:$c_j, j = 1, 2, ..., C$;  Image Prompt Pool:$\mathcal{P}^i$;  Language Prompt Pool:$\mathcal{P}^l$; Domain centroids:$\mathcal{M}$; Number of domain centroids:$N_M$; Number of neighbors for K-NN:$K$;
2: Calculate image feature $f_x = f([x, x_{cls}])$
3: **for** $j = 1, 2, ..., N_M$ **do**
4:    Extract the $j$-th domain centroid $m$ from $\mathcal{M}$
5:    Calculate L1 distance $d_j = \sum |f_x - m|$
6: **end for**
7: Arrange the calculated Euclidean distances $[d_j]_{j=1}^{N_M}$ in a non-descend order
8: Take the first $K$ distances from the sorted list
9: Run K-NN to find the corresponding domain identification $s$
10: Prepend $x$ with image prompts $P_s^i$ for image encoder by $[x, P_s^i, x_{cls}]$
11: Prepare input with language prompts $P_s^l$ for text encoder by $t_j = \left\{ P_s^l, c_j \right\}_{j=1}^{C}$
12: Generate class representation by $g(t_j)$
13: Compute the prediction probability by $p = \dfrac{\exp(\langle f([x, P_s^i, x_{cls}]), g(t_j) \rangle)}{\sum_{k=1}^{C} \exp(\langle f([x, P_s^i, x_{cls}]), g(t_k) \rangle)}$ for classification

---

of 0.1, a batch size of 128, and a cosine annealing scheduler [19]. The number of epochs is different for different datasets, which should be large enough to fit the training sets. Specifically, we use 50 epochs for CDDB, 10 epochs for DomainNet and CORe50. All the training images are resized to $224 \times 224$. We only adopt simple data augmentation as other methods [20, 6], e.g., horizontal flip and random crop.

**Memory overhead.** The number of context vectors for CLIP is 16 and the dimension is 512, and thus the number of parameters of language prompts for each session is $34,816$. The length of image prompts is 10 and the dimension is 768, which has $33,792$ parameters. The dimension of K-Means cluster centers is 512 for S-liPrompts and 768 for S-iPrompts. In our experiments, we use 5 centers for all the three used benchmark datasets, and hence it adds $12,288$ to $18,432$ parameters per session. Accordingly, S-iPrompts only adds $52,224$ parameters to the original pre-trained ViT-B/16 model, leading to a paltry $0.05\%$ total parameter increase. Similarly, S-liPrompts adds $80,896$ parameters per session. Compared with pre-trained CLIP (with ViT-B/16 image encoder), it has a total parameter increase of $0.03\%$.

**Baseline methods.** We implement EWC [9] and LwF [15] with pre-trained ViT-B/16 using the PyCIL[4] toolbox [30]. The hyper-parameters remain the same as those in their original paper. DyTox [2] and L2P [31] are two main baselines in this paper, and we list the parameters below for reference. We use the official implementation[5] of DyTox for its evaluation on CDDB, CORe50 and DomainNet. The original backbone of DyTox is a shallow ConViT [3] (5 self-attention blocks and a task-attention block) without pre-training. For a fair comparison with other methods, we use a pre-trained base-ConViT[6] with first 5 layers as the backbone. The model size of 5 layers base-ConViT (78M) is almost the same as the base-ViT (86M) used in other methods including L2P and ours. We use the learning rate of 0.0001 and 50 epochs for CDDB to get the proper performance. For DomainNet and CORe50, the learning rate is 0.001 and the number of epoch is 20. All the other parts of DyTox remain the same as those of the official implements, and the changes of learning rate and the number of epochs are for getting better performance. We evaluate the official L2P codebase[7] directly on CDDB and DomainNet by tuning its key parameters, i.e., learning rate, trade-off parameter $\lambda$, prompt pool size, $N$ value for key selection. The learning rate for CDDB is tuned to 0.1, and that for DomainNet is 0.8. We use 30 epoch for CDDB and 20 for DomainNet. For both datasets, the prompt pool size is tuned to 20, and the number of key selection is 5. The trade-off parameter $\lambda$ is adjusted to 0.001.

---

[4]`https://github.com/G-U-N/PyCIL`

[5]`https://github.com/arthurdouillard/dytox`

[6]DyTox [2] originally suggests training base-ConViT from scratch. But this case's performance (about 56%) is much lower than that (86.21%) of training from the pre-trained base-ConViT (78M), which is also higher than that (84.20%) of training from the pre-trained base-ViT (86M), on the CDDB-Hard dataset.

[7]L2P [31] merely releases the exemplar-free code at `https://github.com/google-research/l2p`. We tried to adapt it to the cases using exemplars, but they generally work badly, e.g., about 59% on CDDB-Hard.

Table 1: Average accuracies (%) of using random domain selection and our proposed domain selection on CDDB and DomainNet for domain incremental learning. The two domain selection cases are both based on the proposed S-liPrompts.

|  | Random Domain Selection | Proposed Domain Selection | Relative Improvement |
|---|---|---|---|
| CDDB | 80.12 | 88.65 | 8.53 |
| DomainNet | 49.94 | 67.78 | 17.84 |

Table 2: Average accuracies (%) of the originally proposed S-liPrompts (each inference uses one single domain-selected CLIP prediction) and its ensemble version (each inference utilizes the voting strategy on all domain-based CLIP predictions) on the CDDB and DomainNet datasets for domain incremental learning.

|  | Voting for Ensemble | Proposed Domain Selection | Relative Improvement |
|---|---|---|---|
| CDDB | 65.47 | 88.65 | 23.18 |
| DomainNet | 58.85 | 67.78 | 8.93 |

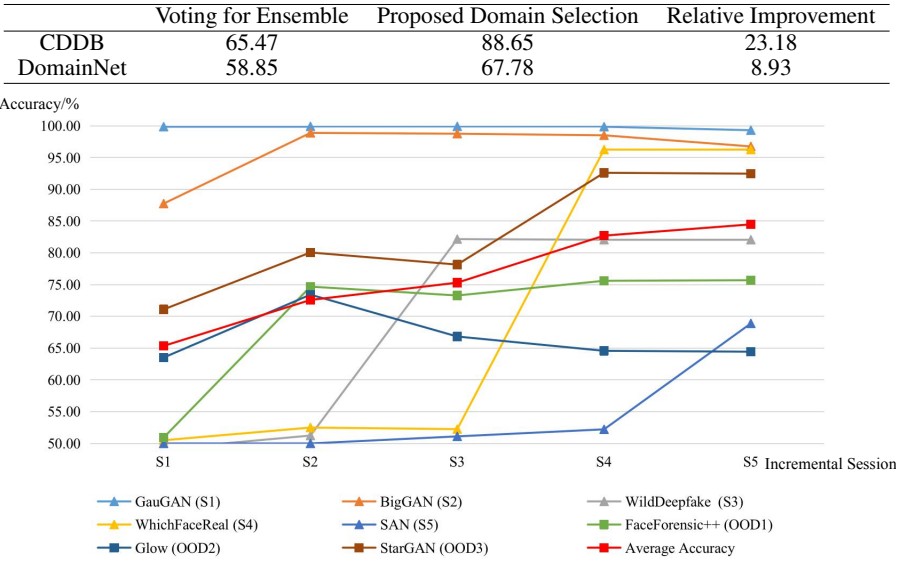

Figure 2: Accuracy curves of the proposed S-liPrompts generalized from five seen CDDB-Hard domains (S1-S5: GauGAN, BigGAN, WildDeepfake, WhichFaceReal, SAN) to three unseen domains (out-of-domain) OOD1-OOD3: FaceForensic++, Glow, StarGAN, which were used by CDDB [12]. The value (S1-S5) along the horizontal direction indicates the S-liPrompts model when trained until the given seen domain data. For example, S3 means that the model was trained along the stream of S1, S2 and S3 sequentially. The curves with different colours indicate the detection performance on the eight seen/unseen domains (S1-S5 and OOD1-OOD3) respectively, and the red one shows the average accuracy.

## 1.3 More Experiments

**Random Domain Selection vs. Proposed Domain Selection.** In addition to the comparison on CDDB (Table 4 in the main paper), we further compare the case of performing random domain selection against that of using our proposed domain selection on DomainNet. Table 1 reports the results on these two datasets. It shows the consistently remarkable relative improvements (by more than 8% on CDDB and about 18% on DomainNet) of the proposed domain selection over the random domain selection. This considerable improvement justifies the necessity of the proposed domain selection, which attributes to the significance of the proposed independent prompting paradigm. The out-of-distribution (OOD) experiment in Table 7 (in the main paper) verifies that the random domain selection (in Table 1) has promising results mainly due to the strong generalization of the proposed language-image prompt learning scheme over the CLIP model. Nevertheless, the results can be further improved significantly using the proposed domain selection (see Table 1), showing the clear superiority of the proposed independent prompting paradigm.

**Domain Ensemble vs. Proposed Domain Selection.** We further study one of the most popular ensemble methods (i.e., voting) on the proposed S-liPrompts. Table 2 summarizes the comparison of the proposed S-liPrompts against the voting-based version in terms of average accuracies on CDDB for deepfake domain incremental learning. The significant relative increases (about 23% on CDDB, 9% on DomainNet) show that the proposed S-liPrompts method favors the proposed domain selection method that chooses the most expertised (i.e., the most related domain-based) prompting model for the inference on each test sample. The excellent performance of the proposed domain-based prompting models verifies the significance of the proposed independent prompting scheme. For the

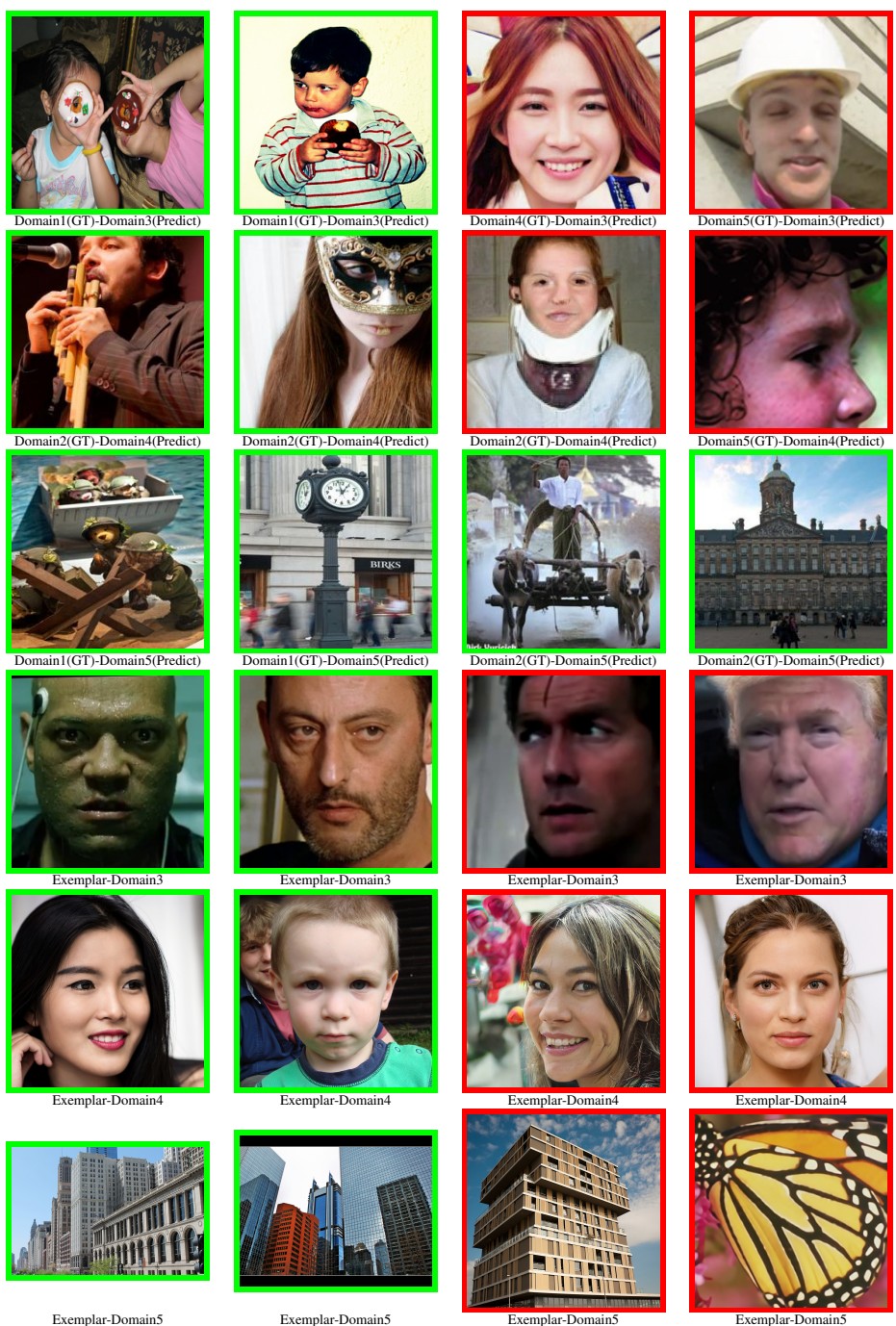

Figure 3: Success examples that are predicted to incorrect domains but are classified correctly in the end using the proposed S-liPrompts on CDDB-Hard. DomainA(GT)-DomainB(Predict): ground truth domain is A, and predicted domain is B. Exemplar-DomainA: exemplars from domain A. Domain1-Domain5: GauGAN, BigGAN, WildDeepfake, WhichFaceReal, SAN. All images with a green boundary are real, and all images with a red boundary are fake.

voting-based inference on each test sample, we first feed all the learned domain-based prompts to the CLIP models one by one resulting in multiple predictions, and we then use the majority voting strategy on the individual results to get the final prediction. The clearly superior performance of the proposed S-liPrompts with domain selection mainly stems from the most expertised model for the given test sample. Except for this model, the rest ones are less expertised for the given test sample, and they might be dominant to corrupt the final prediction when doing the majority voting. In this

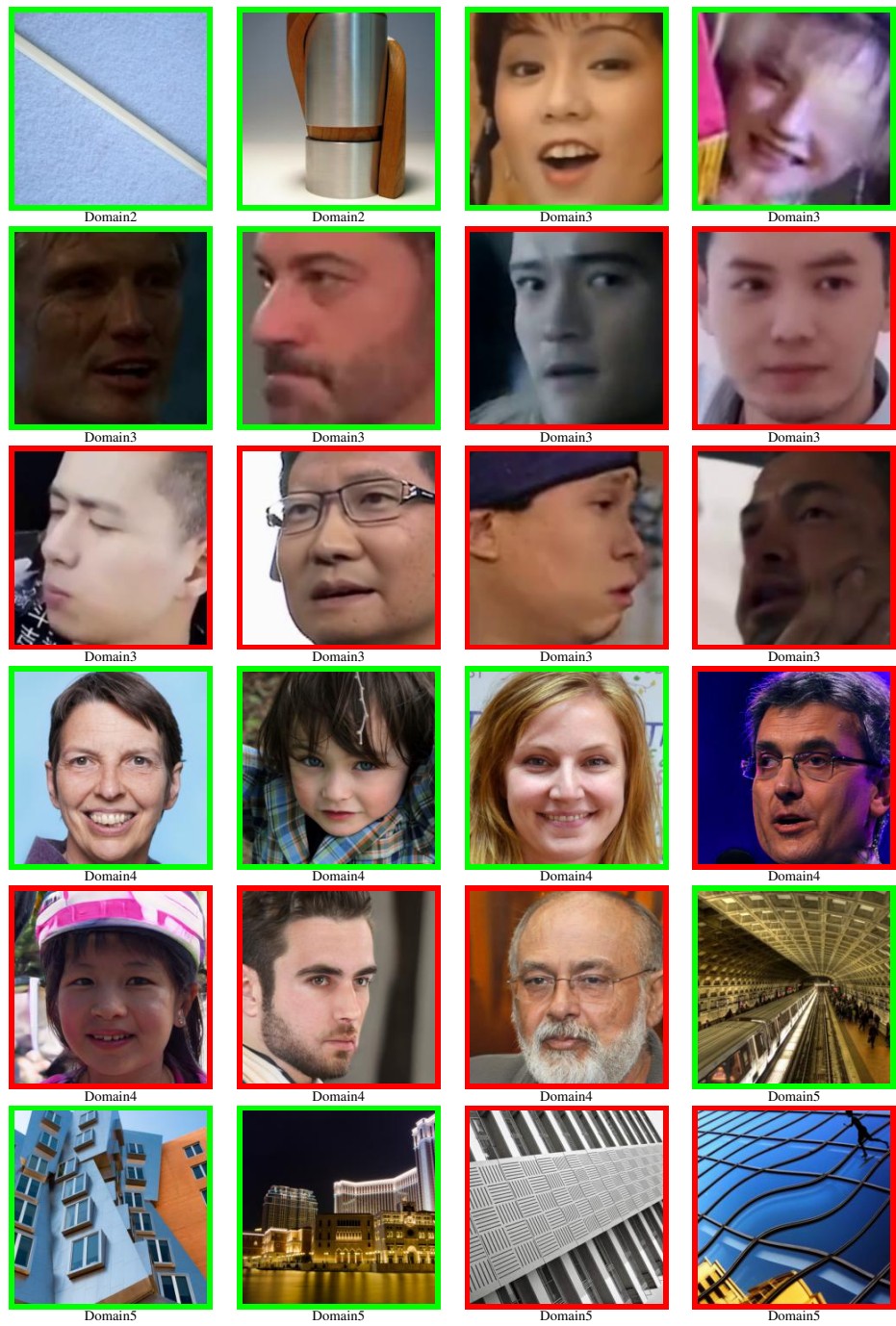

Figure 4: Failure examples that are predicted to correct domains but are classified incorrectly using the proposed S-liPrompts on CDDB-Hard. Domain1-Domain5: GauGAN, BigGAN, WildDeepfake, WhichFaceReal, SAN. All images with a green boundary are real, and all images with a red boundary are fake.

case, the voting scheme could lead to clear performance degradation. Moreover, it requests running all the learned CLIP-based prompting models for the voting on each test sample, and thus it is much more time-consuming than the proposed inference scheme.

**Test on Unseen Domains.** Based on Table 7 in the main paper, Fig.2 additionally shows the detection accuracy curves of the proposed S-liPrompts when applied to three unseen domains (FaceForensic++, Glow and StarGAN). The S-liPrompts models were trained on the five seen domains (GauGAN, BigGAN, WildDeepfake, WhichFaceReal and SAN) from CDDB-Hard sequentially. As studied in

Table 3: Detection accuracies (%) of the proposed S-liPrompts/S-iPrompts and the main competing methods (L2P and DyTox) on the CDDB dataset for deepfake domain incremental learning. **Bold**: best results, Underline: second best results.

|  | GauGAN | BigGAN | WildDeepfake | WhichFaceReal | SAN | Average | Min | Max |
|---|---|---|---|---|---|---|---|---|
| S-liPrompts | 99.30 | 96.75 | 82.06 | 96.25 | 68.89 | **88.65** | **68.89** | **99.30** |
| S-iPrompts | 90.30 | 81.88 | 72.76 | 84.25 | 43.30 | 74.50 | 43.30 | 90.30 |
| L2P | 80.73 | 62.60 | 58.98 | 57.48 | 46.59 | 61.28 | 46.59 | 80.73 |
| DyTox | 48.91 | 50.00 | 59.19 | 50.00 | 50.00 | 51.62 | 48.91 | 59.19 |

Table 4: Precisions (%) of the proposed S-liPrompts/S-iPrompts and the main competing methods (L2P and DyTox) on the used CDDB dataset for deepfake domain incremental learning. Here precision is the ratio tp/(tp + fp) where tp is the number of true positives and fp is the number of false positives. **Bold**: best results, Underline: second best results.

|  | GauGAN | BigGAN | WildDeepfake | WhichFaceReal | SAN | Average | Min | Max |
|---|---|---|---|---|---|---|---|---|
| S-liPrompts | 99.80 | 98.45 | 81.97 | 96.54 | 75.76 | **90.50** | **75.76** | **99.80** |
| S-iPrompts | 90.95 | 83.64 | 73.21 | 84.08 | 43.75 | 75.13 | 43.75 | 90.95 |
| L2P | 83.10 | 77.29 | 65.39 | 57.79 | 66.67 | 70.05 | 57.79 | 83.10 |
| DyTox | 52.99 | 50.00 | 85.86 | 50.00 | 50.00 | 57.77 | 50.00 | 85.86 |

the main paper, the results in Fig.2 demonstrate that the proposed S-liPrompts owns an excellent generalization ability on the unseen domains.

**Successes and Failures.** Fig.3 lists some success cases which are predicted to incorrect domains but are classified correctly in the end using S-liPrompts on CDDB-Hard. It is interesting to see that most of them are predicted to a semantically close domain. For example, as shown in the first two rows in Fig.3, some faces from Domain 1, 2, 4, 5 are predicted to either Domain 3 (WildDeepfaces) or Domain 4 (WhichFaceReal), where all the images are of faces as shown by the exemplars of Domain 3 and Domain 4 (in row 4 and row 5 of Fig.3). In this case, they are very likely to be classified correctly, due to the excellent learning power of S-liPrompts on Domain 3 (WildDeepfaces) and Domain 4 (WhichFaceReal). The same observation can be derived on those images at row 3 in Fig.3. As their architecture details are very similar to those in the exemplars of Domain 5 (SAN), they are assigned to Domain 5 that still leads to the right classification. Fig.4 shows some classical failure cases. As we can see in Table 6 (in the main paper) , the challenging cases are mostly on the domains of WildDeepfake and SAN. For the SAN cases, we can see that some irregular-looking real buildings (e.g., wired structures) are misclassified as fake, while those fake buildings with realistic-looking structures are classified as real, as displayed in Fig.4. For the WildDeepfake domain, there exist a large number of real faces that are often of blurry or occluded or low contrast or non-frontal. These real faces are generally hard for real/fake detection. On the other hand, the fake faces are often of the same quality. This highly increases the challenge for the deepfake detection.

**Feature Spaces of Competing Methods.** Fig.5 uses t-SNE to visualize the resulting feature spaces of pre-trained ViT, CLIP, the two most competing methods DyTox [2] and L2P [31] as well as our S-iPrompts and S-liPrompts on CDDB-Hard. As we discussed in the main paper, dependent learning across domains is very likely to mix up the subspaces of old/new domains. The t-SNE results of L2P and DyTox can verify this to some extent. In contrast, our proposed S-Prompts (especially S-liPrompts) is capable of producing much more separable subspaces. This demonstrates that our independent learning could reach a more favorable feature space where subspaces of different domains are pushed far away from each other. This property could lead to better performances that were reported in the main paper, showing the clear superiority of the proposed independent prompting paradigm over the commonly-respected dependent learning principle.

**Metrics to Measure Separation Degree.** We used t-SNE as it is a popular choice for visualization. Though the significant superiority of our proposed S-liPrompts is shown clearly via t-SNE, it is mainly in terms of domain separation rather than class separation. This phenomenon is mainly from the fact that t-SNE visualization is limited to the 2-dim projection of the original high-dimensional data. Therefore, apart from using t-SNE visualization, we additionally use quantitative metrics to measure the degree of class separation domain by domain. Accordingly, we (re-)collected the classification/detection accuracies in Tables 1 & 2 (main paper), which reflects the average domain-wise accuracies and thus can be a favorable metric for this purpose. Following the related paper [12], we additionally introduce a precision based metric to measure the domain-wise class separation

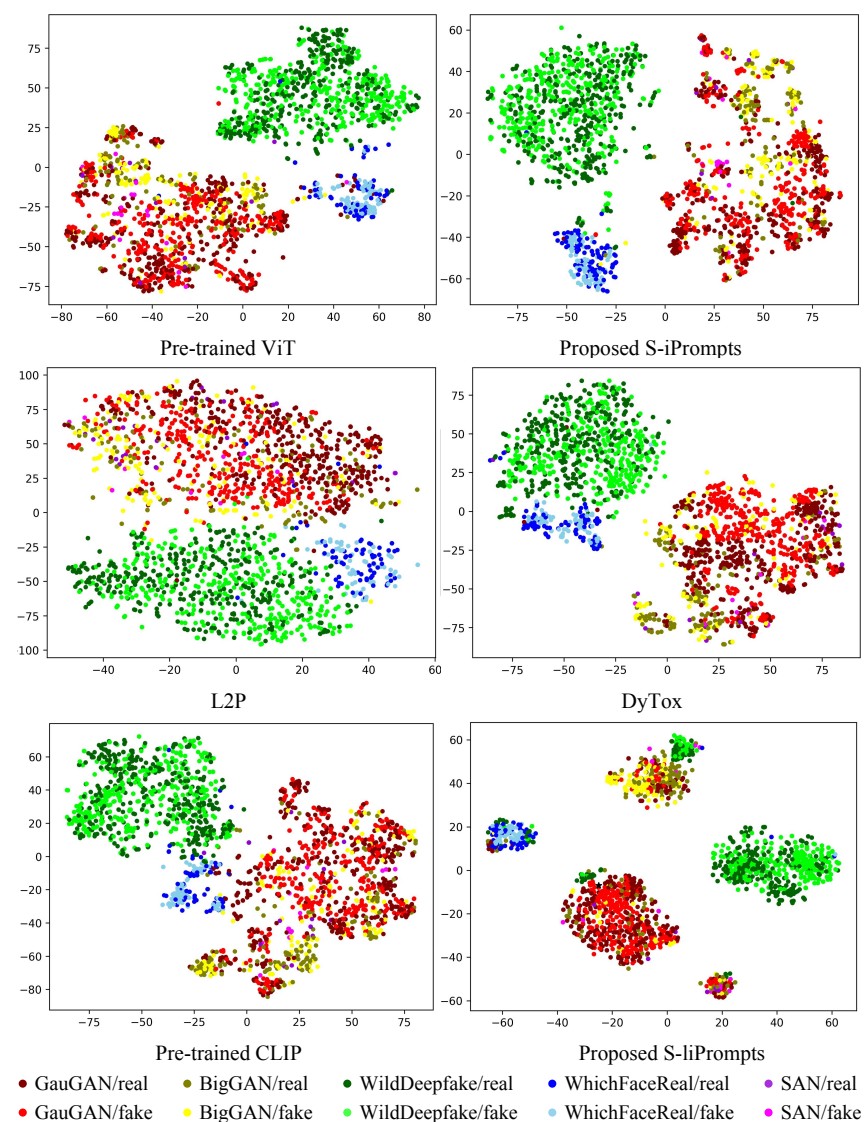

- ● GauGAN/real
- ● BigGAN/real
- ● WildDeepfake/real
- ● WhichFaceReal/real
- ● SAN/real
- ● GauGAN/fake
- ● BigGAN/fake
- ● WildDeepfake/fake
- ● WhichFaceReal/fake
- ● SAN/fake

Figure 5: t-SNE visualization on the resulting feature spaces of pre-trained ViT, proposed S-iPrompts, DyTox [2], L2P [31], pre-trained CLIP, and proposed S-liPrompts on the CDDB-Hard dataset.

Table 5: Memory overheads of the proposed S-iPrompts (ViT-based), S-liPrompts (CLIP-based), the ViT-based prompting methods (DyTox and L2P), and ViT-based non-prompting ones (Others without expansion) on the CDDB dataset. In the setup, there are 5 sessions, each of which includes 2 classes (real and deepfake classes). The average increase corresponds to the parameter increase per session.

|  | DyTox | L2P | Proposed S-iPrompts | Proposed S-liPrompts | Others |
|---|---|---|---|---|---|
| Base Model | 86M | 86M | 86M | 201M | 86M |
| Total Increase | 7.09M | 92.16K | 0.26M | 0.40M | 0 |
| Average Increase | 1.42M | 18.43K | 52.22K | 80.89K | 0 |
| Relative Increase (Increase/Base) | 1.65% | 0.02% | 0.05% | 0.03% | NA |

degree. The results in Table 3 and Table 4 demonstrate the consistent superiority of the proposed S-iPrompts/S-liPrompts methods in terms of the class separation using such various metrics.

**Memory Consumption Comparison.** We further report the memory overheads of the proposed S-iPrompts/S-liPrompts methods (ViT/CLIP-based), the ViT-based prompting methods (DyTox and L2P) as well as those ViT-based non-prompting methods (without expansion). As the non-prompting methods do not expand their architectures for the increasing tasks, their model parameters keep the same for the learning process. Therefore, the main comparison is on the prompting based methods

Table 6: Accuracies (%) of the exemplar-based DyTox using two different backbones (i.e., base-ViT [1] and base-ConViT [3]) on the three benchmarks.

| | DyTox using Pre-trained base-ViT | DyTox using Pre-trained base-ConViT |
|---|---|---|
| CDDB | 84.20 | 86.21 |
| CORe50 | 80.11 | 79.21 |
| DomainNet | 60.83 | 62.94 |

Table 7: Accuracies (%) of the proposed S-iPrompts and the main competing exemplar-free methods (L2P and DyTox) on the three used datasets. Note that CDDB is for continual binary-class classification, and CORe50 and DomainNet are both for continual multi-class classification. DyTox generally fails for the two multi-class classification tasks (CORe50 and DomainNet), as it requests a distillation on examplars for the more challenging balance problem among multiple classes from new/old domains.

| | L2P | DyTox | Proposed S-iPrompts | Relative Improvement |
|---|---|---|---|---|
| CDDB | 61.28 | 51.27 | 74.51 | 13.23 |
| CORe50 | 78.33 | Fails | 83.13 | 4.80 |
| DomainNet | 40.15 | Fails | 50.62 | 10.47 |

including ours. As shown in Table 5, the proposed S-iPrompts increases the model parameters for the domain-specific prompts that are of 52.22K (0.05% relative increase) for each domain (session), and the proposed S-liPrompts's increase is of 80.89K (0.03% relative increase) per session. By comparison, DyTox needs to add one task-attention block over the ViT-based model and dynamically expands the domain-specific tokens (prompts) every session. L2P is like ours directly using ViT-based model without adding any neural network blocks, and it assigns a certain memory to save the pool of used prompts once for the initialization. From the results, we can see that DyTox's increase is the most, and that of L2P is the least. The relative increase ratio of the proposed S-liPrompts/S-iPrompts is very close to that of L2P, while obtaining clearly better performances than the other competing methods.

**Backbones Used by Competing Methods.** As presented in the main paper, to compare fairly, we used the same backbone (i.e., base-ViT [1]) for all the real competitors (except for DyTox that used a more advanced ViT model, i.e., ConViT [3]) as well as the proposed methods (S-iPrompts, S-liPrompts) for all the experiments (Table 1, 2, 3). We reported the results of DyTox with base-ConViT [3] rather than base-ViT [1] in the main paper, since the performance (86.21%) of the former model is better than that (84.20%) of the latter model on CDDB. Hence, we follow the suggestion of the original DyTox paper to use a more advanced ViT model (ConViT) for DyTox in the main paper. Besides, DyTox with base-ViT performs very badly on CDDB with the average accuracy being 48.89%, which is even worse than that of the random guess on CDDB. We further evaluate DyTox with base-ViT on the other two datasets (CORe50 and DomainNet), and we find that the ViT-based exemplar-free DyTox still fails on these two datasets as discovered on the ConViT-based DyTox in the main paper. This is mainly because DyTox requests a distillation on exemplars, which are more demanded to balance the multiple classes from new/old domains on CORe50 and DomainNet, for a promising prompt learning. Therefore, we report the results of the better DyTox (DyTox with base-ConViT) in the main paper, and we here merely report its results using examplars on the three benchmark datasets. The results of Table 6 show that DyTox with base-ConViT generally outperforms (or at least comparable with) DyTox using base-ViT on the three used datasets.

Table 8: Accuracies (%) of ablating components of the proposed S-liPrompts on the CDDB dataset. Note that we use the official implements13 of CLIP to do Zero-shot classification on three benchmarks without any change. Here we use the text templates of ImageNet as their implementation for CLIP (Zero-shot). LIP: Language-Image Prompt, LP: Language Prompt.

| Method | Prompting Scheme | Corresponding Method | CDDB |
|---|---|---|---|
| CLIP (Zero-shot) | handcrafted language prompts | Vanilla CLIP | 49.52 |
| S-liPrompts (fixed LIP Length) | dependent image prompts tuning + dependent language prompts tuning | CLIP + L2P | 64.90 |
| S-liPrompts (fixed LP weights) | independent image prompting + dependent language prompts tuning | CLIP + S-iPrompts | 72.29 |
| S-liPrompts (final, K=5) | independent image prompting + independent language prompting | CLIP+ S-liPrompts | 88.65 |

Table 9: Accuracies (%) of ablating components of the proposed S-liPrompts on the CORe50 and DomainNet datasets. LIP: Language-Image Prompt.

| Method | Prompting Scheme | Corresponding Method | CORe50 | DomainNet |
|---|---|---|---|---|
| S-liPrompts (fixed LIP Length) | dependent image prompts tuning + dependent language prompts tuning | CLIP + L2P | 85.51 | 56.79 |
| S-liPrompts (final, K=5) | independent image prompting + independent language prompting | CLIP+ S-liPrompts | 89.06 | 67.78 |

**Contribution of Independent Image-end Prompting.** Table 7 reports the results that are copied from Table 1, 2, 3 of the main paper. The considerable relative improvement (13% on CDDB, 5% on CORe50, 10% on DomainNet) of the proposed S-iPrompts (independent prompting) over the two main competitors L2P and DyTox (dependent prompting) justifies the significant superiority of the contribution of independent image prompting. We conduct this comparison fairly in the scenario of exemplar-free domain incremental learning (DIL), which is the main aim of our paper, i.e., better data security, privacy and less memory consumption for DIL. In the context of exemplar-free DIL, DyTox collapses on CORe50 and DomainNet, because it requires a distillation on selected exemplars for a better balance among multiple classes from new/old domains on these two datasets, which seems essential for a promising dependent prompting [2]. In addition, we also compared the proposed S-iPrompts against those state-of-the-art exemplar-based methods for a reference. We are delighted to find that S-iPrompts (exemplar-free) outperforms the exemplar-based DyTox clearly (about 4% increase) and the exemplar-based L2P (about 2% improvement) on CORe50, although it is outperformed by the exemplar-based DyTox on CDDB and DomainNet using a large number of exemplars (e.g., 17,250 exemplars on DomainNet). The cost of using large exemplar data is generally not favorable in the real-world scenarios.

**Contribution of Independent Language-Image Prompting.** Table 8 presents the results that are mainly from Table 4 in the main paper, with additional notes in the second and third columns for further clarification. As shown in Table 8, performing CLIP or CLIP+S-liPrompts directly cannot achieve improved performance over S-liPrompts using ViT as a backbone. In contrast, applying the proposed language-image prompting scheme on the CLIP model makes remarkable improvement. From the comparison between S-liPrompts (final, $K = 5$) and S-liPrompts (fixed Language Prompt weights), we can find the contribution of language-image prompting brings an significant improvement of about 16% over the proposed S-iPrompts that is based on the contribution of independent image-end prompting. By comparing S-liPrompts (fixed Image & Language Prompt Length) against S-liPrompts (fixed Language Prompt weights), we can see the considerable superiority (about 8% relative improvement) of S-iPrompts (independent image-end prompting) over L2P-like learning paradigm (dependent image-end prompting), which verifies the significant contribution of independent image-end prompting again. Additionally, the results in Table 1, 2, 3 of the main paper show that our best model S-liPrompts supasses the best of the state-of-the-art exemplar-free methods significantly (30% relative improvement on average) for the three standard DIL benchmark datasets, and even outperforms them clearly (an average increase of 6%) when they use exemplars.

In addition, we evaluate the ablation on CORe50 and DomainNet. Table 9 shows the results. The observation on these two datasets is consistent with that on CDDB. In particular, for DomainNet, the considerable improvement of the final S-liPrompts model demonstrates its clear superiority over the ablated model. In comparison, the improvement on CORe50 is clear but relatively smaller for the CLIP case, while the increase is remarkable when comparing our ViT-based method (S-iPrompts) against ViT-based L2P on CORe50. This might be because CORe50's domain gaps are not as large as those of CDDB and DomainNet (the CORe50 domains are captured by the same Kinect sensor under different backgrounds and lighting [18]). On CORe50, we find most of the evaluated methods have much less forgetting compared to those on CDDB and DomainNet due to the smaller domain gaps of CORe50. In this case, the more powerful language-image prompting scheme might share beneficial knowledge from those similar domains, which does not harm the performance a lot. In contrast, the relative improvements are very remarkable on CDDB (Table 8) and DomainNet (Table 9), showing the significant superiority of the proposed prompting method in the cases with large domain gaps.

### 1.4    More Discussions on Related Works

As presented in the main paper, there exist some expansion-based continual learning methods like [28, 11, 4, 25, 7, 5, 14]. They either dynamically expand the network architectures or reshape their internal structures task by task. This results in the following three issues: 1) Though they generally separate the structure and parameter learning significantly, they still request effective sharing of parameters among tasks. The sharing-driven network tuning inevitably results in a non-trivial trade-off (or so-called tug-of-war) between old task learning and new task learning [24, 10]. 2) They

---

[13]`https://github.com/openai/CLIP/blob/main/notebooks/Prompt_Engineering_for_`
`ImageNet.ipynb`

generally require that the task indexes should be provided for the inference. 3) They often result in a significantly growing number of model parameters for a large number of tasks.

To overcome the second issue of the expansion-based methods, [27, 16] suggest learning a single classifier on a concatenation of all resulting feature maps from different subsets of parameters, which connects all the tasks/domains. Nevertheless, it suffers from the dramatically increased memory overhead issue when handling a long task sequence, which thus requires a complex post-processing for pruning. To overcome this large memory consumption (i.e, the third issue mentioned above), [2] suggests expanding prompts instead over a transformer network. The prompt expansion leads to a small amount of parameter increase. Yet, as discussed in the main paper, like many of traditional methods, the dependent learning across tasks always goes for a tug-of-war (i.e., the first issue mentioned above), where it can only achieve a trade-off between transferring and forgetting, and thus it is almost impossible to achieve the best for every task/domain. By comparison, our paper breaks the commonly-respected learning principle and instead establishes a new independent prompting paradigm to play a win-win game, where it can achieve the best for each domain. In addition, our proposed S-Prompts can address another two issues well with a rough domain identifier and a negligible parameter increase through the proposed prompting paradigm.

There have been also emerging some other transformer-based incremental learning methods like [29, 8, 13], and other incremental prompting methods like [17, 26]. The transformer-based methods without prompting demand much more tuning on the whole network than the prompting methods like [31] and ours. Also, they keep applying traditional dependent learning paradigm to play the tug-of-war game. The prompting method [17] suggests learning episodic memory prompts to explicitly carry previously learned knowledge to subsequent tasks for class-incremental event detection. However, it still follows the conventional learning paradigm to perform experience replay and knowledge distillation for the trade-off between transfer and interference.