# OpenReview forum: "S-Prompts Learning with Pre-trained Transformers: An Occam’s Razor for Domain Incremental Learning"
_NeurIPS.cc/2022/Conference — NeurIPS 2022 Accept_

### Official Review · Reviewer_CnBo · 2022-07-08

**Rating:** 5
**Confidence:** 5
**Soundness:** 3 good
**Presentation:** 3 good
**Contribution:** 2 fair

**Summary:**

This paper proposed S-Prompting and two concrete approaches to highly reduce the forgetting degree in continual learning scenarios, i.e., domain incremental learning (DIL).

The main idea is to learn independent prompts across domains with pre-trained transformers.

The independent prompt can achieve the best for each domain.

The learning method derives an image prompt learning approach and a brand-new language-image prompt learning approach.

The methods outperformed all three standard DIL tasks.


**Questions:**

In Tables 1 and 2, for fair comparisons with others, the architecture should be the same as the base-ViT model.
In the appendix, as the authors state, the number of parameters -ConViT(78M) and base-ViT(86M) are almost similar. However, I think if the structures are one another, the representations also differ.
I would recommend the authors compare the performances under the same architecture such as the base-ViT(86M).


**Limitations:**

The authors discuss the limitation of the S-prompts properly in the script.


**Strengths And Weaknesses:**

(+) Compared with prior works: L2P and DyTox, the proposed S-Prompts learn the tasks independently.

(+) The S-prompts are technically sound and seem to be well supported by experimental results on the three standard Domain-IL benchmark datasets, and outperformed others.

(-) In Tables 1 and 2, for fair comparisons with others, the S-prompts compared with others under a little bit different architecture.

(-) S-prompts (S-liPrompts) seems strongly dependent on CLIP (a pre-trained Image-text mapping function), which decreases the main novelty.

---

> ### Author Response · Authors · 2022-08-02
> **Response2 to Reviewer CnBo**
>
> ### **Q2. S-prompts (S-liPrompts) seems strongly dependent on CLIP**
>
> In the paper, we make two main technical contributions: (1) independent image-end prompt learning paradigm, and (2) language-image prompting scheme with the pre-trained CLIP model. While the proposed independent prompting paradigm can be applied to any pre-trained transformer based models, we exploited two concrete methods (S-iPrompts, S-liPrompts) based on ViT and CLIP respectively in this paper (Line 74-76 of the main paper).
>
> Table R12 reports the results that are from Table 1, 2, 3 of the main paper. The considerable relative improvement (13% on CDDB, 5% on CORe50, 10% on DomainNet) of the proposed S-iPrompts (independent prompting) over the two main competitors L2P and DyTox (dependent prompting) justifies the significant superiority of the contribution (1). We conduct this comparison fairly in the scenario of exemplar-free domain incremental learning (DIL), which is the main aim of our paper, i.e., better data security, privacy and less memory consumption (Line 34-36 of the main paper, as acknowledged by Reviewer GPZj). In the context of exemplar-free DIL, DyTox collapses on CORe50 and DomainNet, because it requires a distillation on selected exemplars for a better balance among multiple classes from new/old domains on these two datasets, which is essential for a promising dependent prompting [14] (Footnote 4 in the main paper). In addition, we also compared the proposed S-iPrompts against those state-of-the-art exemplar-based methods for a reference. We are delighted to find that S-iPrompts (exemplar-free) outperforms the exemplar-based DyTox clearly (about 4% increase) and the exemplar-based L2P (about 2% improvement) on CORe50, although it is outperformed by the exemplar-based DyTox on CDDB and DomainNet using a large number of exemplars (e.g., 17,250 exemplars totally on DomainNet). The use of large exemplar data is generally not favorable in the real-world scenarios.
>
>
> ---
> Table R13: (results are from Table 4 in the main paper). Accuracies (%) of ablating components of the proposed S-liPrompts on the CDDB datasets. Note that we use the official implements [1] of CLIP to do Zero-shot classification on three benchmarks without any change. Here we use the text templates of ImageNet as their implementation [1] for CLIP (Zero-shot).
> [1] https://github.com/openai/CLIP/blob/main/notebooks/Prompt_Engineering_for_ImageNet.ipynb
>
> | Method                                             | Prompting Scheme | Corresponding Method              | CDDB   |
> |----------------------------------------------------|---------------------------------------------------------------------|--------------------|--------|
> | CLIP (Zero-shot)                                   | Handcrafted language prompts                                        | Vanilla CLIP       | 49.52  |
> | S-liPrompts (fixed Image & Language Prompt length) | dependent image prompts tuning + dependent language prompts tuning  | CLIP + L2P         | 64.90  |
> | S-liPrompts (fixed Language Prompt weights)        | independent image prompting + dependent language prompts tuning     | CLIP + S-iPrompts  | 72.29  |
> |  S-liPrompts (final, K=5)                          | independent image prompting + independent language prompting        | CLIP+ S-liPrompts  |  88.65 |
> ---
>
> Table R13 presents the results that are from Table 4 in the main paper, with additional notes in the second and third columns for further clarification. As shown in Table R13, performing CLIP or CLIP+S-liPrompts directly cannot achieve improved performance over S-liPrompts using ViT as a backbone. In contrast, applying the proposed language-image prompting scheme on the CLIP model, i.e, the contribution (2), makes remarkable improvement. From the comparison between S-liPrompts (final, K=5) and S-liPrompts (fixed Language Prompt weights), we can find the contribution (2) brings an significant improvement of about 16% over the proposed S-iPrompts that is based on the contribution (1). By comparing S-liPrompts (fixed Image & Language Prompt length) against S-liPrompts (fixed Language Prompt weights), we can see the considerable superiority (about 8% relative improvement) of S-iPrompts (independent image-end prompting) over L2P-like learning paradigm (dependent image-end prompting), which verifies the significance of the contribution (1) again. Additionally, the results in Table 1, 2, 3 of the main paper show that our best model S-liPrompts, i.e., contribution (1) + contribution (2), supasses the best of the state-of-the-art exemplar-free methods significantly (30% relative improvement on average) for the three standard DIL benchmark datasets, and even outperforms them clearly (an average increase of 6%) when they use exemplars (Line 89-91).
>
> In conclusion, the proposed S-prompts is only partly dependent on CLIP. We added such clarification in the revised supp. material.

---

> > ### Comment · Reviewer_CnBo · 2022-08-09
> > **On ablation studies of S-liPrompts**
> >
> > Thank you for your rebuttal on more ablation studies.
> >
> > Table R13 provides the effectiveness of S-liPrompts.
> >
> > I think CLIP + L2P on CORe50 will be more effective in showing the novelty of the proposed methods if possible.

---

> > > ### Author Response · Authors · 2022-08-09
> > > **Reponse4 to Reviewer CnBo**
> > >
> > >
> > > Thank you for raising the further suggestion. Following this suggestion, we additionally evaluate the mentioned ablation on CORe50 and DomainNet. Table R14 shows the results. The observation on these two datasets is consistent with that on CDDB. In particular, for DomainNet, the considerable improvement of the final S-liPrompts model demonstrates its clear superiority over the ablated model. In comparison, the improvement on CORe50 is clear but relatively smaller for the CLIP case, while the increase is remarkable when comparing our ViT-based method (S-iPrompts) against ViT-based L2P on CORe50. This might be because CORe50’s domain gaps are not as large as those of CDDB and DomainNet (the CORe50 domains are captured by the same Kinect sensor under different backgrounds and lighting [38]). On CORe50, we find most of the evaluated methods have much less forgetting compared to those on CDDB and DomainNet due to the smaller domain gaps of CORe50. In this case, the more powerful language-image prompting scheme might share beneficial knowledge from those similar domains, which does not harm the performance a lot. In contrast, the relative improvements are very remarkable on CDDB (Table R13) and DomainNet (Table R14), showing the significant superiority of the proposed prompting method in the cases with large domain gaps. We will further include such additional experiments and the discussion into the revised supp. material.
> > >
> > > ---
> > > Table R14:  Accuracies (%) of ablating the components of the proposed S-liPrompts on the CORe50 and DomainNet datasets.
> > > | Method                                             | Prompting Scheme                                                    | Corresponding Method | CORe50    | DomainNet |
> > > |----------------------------------------------------|---------------------------------------------------------------------|----------------------|-----------|-----------|
> > > | S-liPrompts (fixed Image & Language Prompt length) | dependent image prompts tuning + dependent language prompts tuning  | CLIP + L2P           | 85.51     | 56.79     |
> > > | **S-liPrompts (final, K=5)**                     | independent image prompting + independent language prompting        | CLIP+ S-liPrompts    | **89.06** | **67.78** |
> > > ---

---

> ### Author Response · Authors · 2022-08-02
> **Response1 to Reviewer CnBo**
>
> We thank Reviewer CnBo for the constructive and valuable feedback, which encourages us to further improve this paper.
>
> ### **Q1. The same architecture for fair comparison**
>
> As presented in Line 236-241 of the main paper, to compare fairly, we used the same backbone (i.e., base-ViT [13]) for all the real competitors (except for DyTox that used a more advanced ViT model, i.e., ConViT [15]) as well as the proposed methods (S-iPrompts, S-liPrompts) for all the experiments (Table 1, 2, 3). In Footnote 3 of the supp. material, we additionally state that we reported the results of DyTox with base-ConViT [15] rather than base-ViT, since the performance (86.21%) of the former model is better than that (84.20%) of the latter model on CDDB. Hence, we follow the suggestion of the original DyTox paper to use a more advanced ViT model for DyTox in the main paper. Following your suggestion, we further evaluate DyTox with base-ViT on the other two datasets (CORe50 and DomainNet), and we find that the ViT-based exemplar-free DyTox still fails on these two datasets as discovered on the ConViT-based DyTox in the main paper. This is mainly because DyTox requests a distillation on exemplars, which are more demanded to balance the multiple classes from new/old domains on CORe50 and DomainNet, for a promising prompt learning [14] (Footnote 4 in the main paper). Therefore, we here report the results of the exemplar-based DyTox. The results of Table R11 show that DyTox with base-ConViT generally outperforms (or at least comparable with) DyTox using base-ViT on the three used datasets. Besides, CaSSLe is a self-supervised incremental learning method, and replacing supervised trained weights trained on ImageNet is not suitable. Hence, we use its officially released backbone. We added such clarification in the revised main paper and the supp. material.
>
> ---
> Table R11. Accuracies (%) of the exemplar-based DyTox using two different backbones (i.e., base-ViT [13] and base-ConViT [15]) on the three used benchmark datasets.
> |           | DyTox using Pre-trained base-ViT | DyTox using Pre-trained base-ConViT (reported in the main paper) |
> |-----------|----------------------------------|------------------------------------------------------------------|
> | CDDB      | 84.20                            | 86.21                                                            |
> | CORe50    | 80.11                            | 79.21                                                            |
> | DomainNet | 60.83                            | 62.94                                                            |
> ---
>
>
>
>
> ---
> Table R12. Accuracies (%) of the proposed S-iPrompts and the main competing exemplar-free methods (L2P and DyTox) on the three used datasets. Note that CDDB is for continual binary-class classification, while CORe50 and DomainNet are both for continual multi-class classification. DyTox generally fails for the two multi-class classification tasks (CORe50 and DomainNet), as it requests a distillation on examplars for the more challenging balance problem among multiple classes from new/old domains.
> |           | L2P   | DyTox | Proposed S-iPrompts | Relative Improvement |
> |-----------|-------|-------|---------------------|----------------------|
> | CDDB      | 61.28 | 51.27 | 74.51               | 13                  |
> | CORe50    | 78.33 | Fails | 83.13               | 5                  |
> | DomainNet | 40.15 | Fails | 50.62               | 10                  |
> ---

---

> > ### Comment · Reviewer_CnBo · 2022-08-09
> > **The same backbone on CDDB.**
> >
> > Thank you for your rebuttal. Now, I understand the status of the experiments.
> >
> > Regarding the backbone issues on CDDB, the authors used ConViT[15] (a more advanced one).
> >
> > I would like to know why the authors didn't report the performance of the based-ViT on CDDB for fair comparisons.

---

> > > ### Author Response · Authors · 2022-08-09
> > > **Reponse3 to Reviewer CnBo**
> > >
> > >
> > > Thanks for making the further comment. The reason why we did not provide the performance of the base-ViT for DyTox on CDDB in the main paper is three-fold. Firstly, the original DyTox paper [14] suggests using a more advanced ViT (i.e., base-ConViT [15]), and thus we directly evaluate its officially implemented model that uses base-ConViT as the backbone to show its full potential for the three standard domain incremental learning tasks in the paper. Secondly, when using exemplars, we find that the performance (86.21%) of the DyTox+base-ConViT [15] is superior to that (84.20%) of the DyTox+base-ViT [13] on CDDB, as presented in Footnote 3 of the original supp. material paper. The similar observation on the other two datasets can be obtained from Table R11. This justifies that ConViT is a better choice for the exemplar-based DyTox. Thirdly, the paper mainly focuses on the comparison to the exemplar-free methods. It is worth noting that the exemplar-free DyTox+base-ViT fails to work and almost collapses (with the average accuracy being 48.89% that is even worse than that of the random guess) on CDDB. By comparison, DyTox+base-ConViT works more normally on CDDB, while its performance (51.27%) is slightly better than that of the random guess for deepfake detection. The inferior performance of the exemplar-free DyTox is mainly due to its strong demand on the use of exemplars for a distillation, which is designed to balance different classes from new/old domains in the context of the DyTox framework. Considering the relatively better performance of DyTox+base-ConViT, we choose to report its results on CDDB for both the exemplar-based and exemplar-free cases. Although DyTox uses a more advanced backbone (i.e., base-ConViT), it is still outperformed by our proposed methods.
> > >
> > > Following your good comment, we will further add this clarification in the revised paper.

---

### Official Review · Reviewer_WPXn · 2022-07-11

**Rating:** 4
**Confidence:** 3
**Soundness:** 2 fair
**Presentation:** 2 fair
**Contribution:** 2 fair

**Summary:**

This paper proposes to address DIL by (1) learning a pool of prompts and classifier heads (one for each domain) for ViT and CLIP (2) inferring domain label based on K-means and KNN.

**Questions:**

Can the authors clarify or disentangle the contributions from (1) CLIP as a backbone and (2) learning independent prompts for each domain?

**Limitations:**

See weaknesses and questions above.

**Strengths And Weaknesses:**

Strengths
1. This paper is written clearly and generally easy to follow.
2. The proposed idea is simple.

Weaknesses
1. I’m not sure whether the performance improvement over baselines (especially L2P and DyTox) are due to the proposed method or the usage of CLIP. The comparisons based on the same backbone, ViT (corresponding to S-iPrompts), don’t show a clear improvement over the two prompt-based baselines. And the significant improvement comes from the CLIP-based models.
2. Related to the previous point, the good performance of random domain selection (Table 4) seems to imply that much of the performance is due to a simple prompt + backbone transformer.
3. Minors: there’re typos and grammatical errors and they should be corrected. e.g., cheep prompts -> cheap prompts?

---

> ### Author Response · Authors · 2022-08-02
> **Response2 to Reviewer WPXn**
>
>
>
> ---
> Table R8: (results are from Table 4 in the major paper). Accuracies (%) of ablating components of the proposed S-liPrompts on the CDDB dataset. Note that we use the official implements [1] of CLIP to do zero-shot classification on three benchmarks without any change. Here we use the text templates of ImageNet as their implementation [1] for CLIP (Zero-shot).
> [1] https://github.com/openai/CLIP/blob/main/notebooks/Prompt_Engineering_for_ImageNet.ipynb
>
> | Method                                             | Prompting Scheme | Corresponding Method                | CDDB   |
> |----------------------------------------------------|---------------------------------------------------------------------|---------------------|--------|
> | CLIP (Zero-shot)                                   | handcrafted language prompts                                        | Vanilla CLIP        | 49.52  |
> | S-liPrompts (fixed Image & Language Prompt length) | dependent image prompts tuning + dependent language prompts tuning  | CLIP + L2P          | 64.90  |
> | S-liPrompts (fixed Language Prompt weights)        | independent image prompting + dependent language prompts tuning     | CLIP + S-iPrompts   | 72.29  |
> |  S-liPrompts (final, K=5)                          | independent image prompting + independent language prompting        | CLIP + S-liPrompts  |  88.65 |
>
> ---
>
>
>
>
> ### **Q2: Good performance of random domain selection seems to imply that much of the performance is due to a simple prompt + backbone transformer**
>
>
> ---
> Table R9. Average accuracies (%) of using random domain selection and our proposed domain selection on CDDB and DomainNet for domain incremental learning. The two domain selection cases are both based on the proposed S-liPrompts.
> |           | Random Domain Selection | Proposed  Domain Selection  | Relative Improvement |
> |-----------|-------------------------|-----------------------------|----------------------|
> | CDDB      | 80.12                   | 88.65                       | 8                    |
> | DomainNet | 49.94                   | 67.78                       | 18                   |
> ---
> ---
> Table R10: Out-of-distribution (OOD) experiments (supp. material paper Table2). Accuracies of the application of the trained S-liPrompts on S1-S5 to out-of-domains OOD1-OOD3 that are 3 unseen domains used in CDDB. S1-S5: GauGAN, BigGAN, WildDeepfake, WhichFaceReal, SAN, which are used to train S-liPrompts on the CDDB dataset. OOD1-OOD3: FaceForensic++, Glow, StarGAN, which are not used to train S-liPrompts.
> |      | OOD1  | OOD2  | OOD3  | Avg   |
> |------|-------|-------|-------|-------|
> | CDDB | 75.69 | 64.44 | 92.46 | 77.53 |
> ---
>
> Apart from the study on CDDB (Table 4 in the main paper), we further compare the case of using random domain selection against that of performing our proposed domain selection on DomainNet. As shown in Table R9, the performance of the random selection is actually not good enough. In particular, it is outperformed by the proposed domain selection significantly, with relative degradation being more than **8%** on CDDB and about **18%** on DomainNet. This considerable improvement demonstrates the necessity of the proposed domain selection, which attributes to the significance of the proposed independent prompting paradigm. In Table R10, the out-of-distribution (OOD) experiment (results from the supp. material Table 2) justifies that the random domain selection (in Table R9) has promising results, which is mainly due to the strong generalization of the proposed language-image prompt learning scheme over the CLIP model. Nevertheless, the results can be further improved remarkably using the proposed domain selection (Table R9). This shows the clear contribution of the proposed independent prompting paradigm. We included such a study on the necessity of performing domain identification/selection in the revised supp. material.
>
> ### **Q3: Typos and grammatical typos**
>
> Thank you for your suggestion. We carefully corrected the typos and grammatical errors throughout the paper.

---

> ### Author Response · Authors · 2022-08-02
> **Response1 to Reviewer WPXn**
>
> We thank Reviewer WPXn for the thorough and constructive feedback. The feedback enables us to further strengthen the paper.
>
> ### **Q1: The contributions from (1) the proposed learning independent image-end prompts for each domain, and (2) using CLIP as backbone with the suggested independent language-image prompting scheme?**
>
>
> ---
> Table R7. Accuracies (%) of the proposed S-iPrompts and the main competing exemplar-free methods (L2P and DyTox) on the three used datasets. Note that CDDB is for continual binary-class classification, while CORe50 and DomainNet are both for continual multi-class classification. DyTox generally fails for the two multi-class classification tasks (CORe50 and DomainNet), as it requests a distillation on examplars for the more challenging balance problem among multiple classes from new/old domains.
> |           | L2P   | DyTox | Proposed S-iPrompts | Relative Improvement |
> |-----------|-------|-------|---------------------|----------------------|
> | CDDB      | 61.28 | 51.27 | 74.51               | 13                   |
> | CORe50    | 78.33 | Fails | 83.13               | 5                    |
> | DomainNet | 40.15 | Fails | 50.62               | 10                   |
> ---
>
> As shown in Table R7 where the results are mainly from Table 1, 2, 3 of the main paper, the remarkable relative improvement (**13%** on CDDB, **5%** on CORe50, **10%** on DomainNet) of the proposed S-iPrompts (independent prompting) over the two main competitors L2P and DyTox (dependent prompting) demonstrates the significant superiority of the contribution (1). This comparison is conducted fairly in the context of exemplar-free domain incremental learning, which is the focus of our paper with the aim being for better data security, privacy and less memory consumption (Line 34-36 of the main paper, as acknowledged by Reviewer GPZj). Note that in the exemplar-free setup, DyTox collapses on CORe50 and DomainNet, since it requests a distillation on exemplars to balance the multiple classes from new/old domains for promising dependent prompting [14] (Footnote 4 in the main paper). For a reference, we also compared the proposed S-iPrompts against those exemplar-based methods. We are delighted to find that S-iPrompts (exemplar-free) surpasses the exemplar-based DyTox clearly (about 4% increase) and the exemplar-based L2P (about 2% improvement) on CORe50, while it is outperformed by the exemplar-based DyTox on CDDB and DomainNet using a large number of exemplars (e.g., 17,250 exemplars totally on DomainNet), which is generally not favorable in the real-world scenarios.
>
> Table R8 presents the results that are from Table 4 in the main paper, with additional notes in the second and third columns for further clarification. The results in Table R8 show that applying CLIP or CLIP+S-liPrompts directly cannot obtain improved performance over S-liPrompts that uses ViT as a backbone. Instead, performing the proposed language-image prompting scheme on the CLIP model, i.e, the contribution (2), brings significant improvement. In particular, by comparing S-liPrompts (final, K=5) against S-liPrompts (fixed Language Prompt weights), we can see the contribution (2) makes a considerable improvement of about **16%** over the S-iPrompts (with the contribution (1)). By comparing S-liPrompts (fixed Image & Language Prompt length) with S-liPrompts (fixed Language Prompt weights), we can discover the clear superiority (about **8%** increase) of S-iPrompts (independent image-end prompting) over the L2P-like learning paradigm (dependent image-end prompting), justifying the significance of the contribution (1) again. In addition, from Table 1, 2, 3 of the main paper, we can see that our best model S-liPrompts, i.e., contribution (1) + contribution (2),  outperforms the best of the state-of-the-art exemplar-free methods significantly (an average of **30%** relative improvement) for the three standard DIL benchmark datasets, and even suparsses them relatively by an average of **6%** when they use exemplars (Line 89-91).
>
> In summary, the contribution (1) and contribution (2) are both significant. For instance, they contribute relative improvement of about 13% and 16% respectively over the competing methods on the CDDB dataset. We added such clarification in the revised supp. material.

---

### Official Review · Reviewer_GPZj · 2022-07-11

**Rating:** 6
**Confidence:** 3
**Soundness:** 3 good
**Presentation:** 4 excellent
**Contribution:** 3 good

**Summary:**

The submission focus on the catastrophic forgetting problem in domain-incremental learning, in which a model needs to learn from different domains sequentially. To achieve the goal in an exemplar-free way, the authors design an S-Prompting mechanism. A new trainable prompt is introduced every time a new domain appears. The trained prompt is then added to a prompt repository. At inference time, when new data comes, the prompt of the corresponding domain is retrieved and prepend to the input of ViT or CLIP. Extensive experiments are conducted and the best S-Prompts surpasses not only exemplar-free methods but also some exemplar-based method.

**Questions:**

1. I could that see there is an analysis in Section 4 and supplementary materials about the overheads of the proposed methods compared to its backbone, but there is no information about the memory overheads of the baselines. I would suggest listing the parameter scale of the baseline methods as well.

2. What do you mean by ``dynamic classifers'' in line 246 and line 272? The term never appears in the approach section.

**Limitations:**

The limitations are adequately discussed in the submission.

**Strengths And Weaknesses:**

Strengths:

1. The proposed approach is clear and elegant. It uses prompt to store the domain-related feature while keeping the backbone model unchanged though it is not the first time that prompt method is used for continual learning. Besides, no additional space is required to store the exemplar.

2. The experiment is extensive and persuasive. Previous methods are properly considered. The performance is impressive, especially when compared with exemplar-based methods.

3. The writing is clear and easy to follow.

Weaknesses:

No serious weaknesses found.

---

> ### Author Response · Authors · 2022-08-02
> **Response to Reviewer GPZj**
>
> We thank Reviewer GPZj for the thorough and constructive feedback that helps us to further improve the paper. We are glad that Reviewer GPZj finds our paper is clear and elegant. We also thank Reviewer GPZj for stating that the paper has no serious weaknesses.
>
> ### **Q1: Memory consumption comparison**
>
>
> ---
> Table R6. Memory overheads of the proposed S-iPrompts (ViT-based), S-liPrompts (CLIP-based), the ViT-based prompting methods (DyTox and L2P), and ViT-based non-prompting ones (Others without expansion) on the CDDB dataset. In the setup, there are 5 sessions, each of which includes 2 classes (real and deepfake classes). The average increase corresponds to the parameter increase per session.
>
> |                                   | DyTox | L2P    | S-iPrompts | S-liPrompts | Others |
> |-----------------------------------|-------|--------|------------|-------------|--------|
> | Base Model                        | 86M   | 86M    | 86M        | 201M        | 86M    |
> | Total Increase                    | 7.09M | 92.16K | 0.26M      | 0.40M       | 0      |
> | Average Increase                  | 1.42M | 18.43K | 52.22K     | 80.89K      | 0      |
> | Relative Increase (Increase/Base) | 1.65% | 0.02%  | 0.05%      | 0.03%       | NA     |
> ---
>
> We follow your suggestion to further report the memory overheads of the proposed S-iPrompts/S-liPrompts methods (ViT/CLIP-based), the ViT-based prompting methods (DyTox and L2P) as well as those ViT-based non-prompting methods (without expansion). As the non-prompting methods do not expand their architectures for the increasing tasks, their model parameters keep the same for the learning process. Therefore, the main comparison is on the prompting based methods including ours. As shown in Table R6 and Line 36-44 (the supp. material), the proposed S-iPrompts increases the model parameters for the domain-specific prompts that are of 52.22K (0.05% relative increase) for each domain (session), and the proposed S-liPrompts’s increase is of 80.89K (0.03% relative increase) per session. By comparison, DyTox needs to add one task-attention block over the ViT-based model and dynamically expands the domain-specific tokens (prompts) every session. L2P is like ours directly using ViT-based model without adding any neural network blocks, and it assigns a certain memory to save the pool of used prompts once for the initialization. From the results, we can see that DyTox’s increase is the most, and that of L2P is the least. The relative increase ratio of the proposed S-liPrompts/S-iPrompts is very close to that of L2P, while obtaining clearly better performances than the other competing methods. We added this table as well as such discussion in the revised supp. material.
>
>
> ### **Q2: Meaning of 'dynamic classifiers'**
>
>
> We originally used “dynamic classifiers” to represent the growing pool of the language model based classifiers in the context of CLIP. As shown in Line 213 and Line 246 of the revised main paper, we used “the growing pool of domain-based classifiers” instead to avoid confusion.

---

### Official Review · Reviewer_Lxav · 2022-07-18

**Rating:** 8
**Confidence:** 4
**Soundness:** 3 good
**Presentation:** 4 excellent
**Contribution:** 4 excellent

**Summary:**

This paper proposes to tackle the problem of domain incremental learning (in which its setting is distinct from the problem of class-incremental learning. In domain incremental learning, different domain data arrive in a sequence, one domain per incremental phase, while these domains share the same group of object/image classes), in which the domain indexes are not given during the inference (therefore dissimilar to the setting of task-incremental learning). As other incremental learning scenarios, the catastrophic forgetting is one of the biggest challenges in domain incremental learning, where the domain gap is main reason behind to cause the forgetting (i.e. while learning to classify new domain data, the model's ability on classifying old domain data could deteriorate). In order to tackle such catastrophic getting issue, instead of additionally maintaining portion of old domain data as what replay-based approaches do (under the consideration for better data security and privacy), the proposed method learns a set of prompts over (pretrained and fixed) transformers where the domain-specific knowledge is stored by a prompt pool, in which during the inference the classification is done by firstly identifying the domain ID (via K-NN to search for the nearest domain centroids obtained by K-means on training data to the test image feature) then feeding the corresponding domain-specific prompt and the image tokens to the transformer to perform the classification via the corresponding domain-specific classifier. With the fact that the proposed method is able to not only achieve image prompt learning (via ViT) but also a brand-new language-image prompt learning (via CLIP), the extensive experiments are conducted to show the superior performance of the proposed method against various baselines (including exemplar-free and replay-based incremental-learning methods, self-supervised learning method, and the recently published prompting methods) on several datasets.

**Questions:**

Overall I pretty enjoy reading the whole paper and like the novel idea of learning domain-specific prompts in a domain-independent way (which strike a much better balance between alleviating the forgetting and learning to recognize new domain data) as well as its applicability on pretrained and fixed image/language-image transformers. The questions that I would encourage the authors to address in the rebuttal, for further strengthening the paper, are the ones listed in cons, including the potential contradiction between the domain-specific prompts and the robustness to erroneous domain identification, and the insignificant separation between different domains.

**Limitations:**

there is no potential negative societal impact

**Strengths And Weaknesses:**

Pros:
+ The proposed method is built upon pretrained (image or even language-image) transformers (without any fine-tuning needed) with advancing the existing prompting methods of continual learning on a novel idea: although existing methods also aim to learn the domain-specific prompts, their learning is dependent across domains thus potentially leading to less separation for the classes from different domains, while the proposed method particularly drives the prompt learning independently across domains to achieve the best for each domain. Experimental results as well as ablation study will verify the contribution of such novel idea.
+ The proposed prompt learning (with only little overhead needed to store the domain-specific prompt) surpasses not only other prompting methods but even the replay-based incremental learning approaches (where the exemplars of old domain data are explicitly stored in the replay buffer and used during the learning for new domain data to alleviate the catastrophic forgetting) and the expansion-based incremental learning approaches (e.g. DER++, where different feature extractors are adopted per domain) , which again demonstrates the contribution of the novel idea behind the proposed method.

Cons:
- Though the authors do provide additional experiments to demonstrate that the random assigning domain ID or having errors on domain identification (i.e. to firstly recognize which domain that the test data belongs to) would not have huge impact on the final classification performance, such argument seems to be contradictive to the idea of having domain-specific prompts and learning them in a domain-independent way. In other words, if using the wrong domain-specific prompt can still provide the slightly worse but comparable performance, then perhaps it could be unnecessary to perform domain identification. Instead, directly having the ensemble over the results produced by using each of the domain-specific would potentially contribute the best performance? The authors are highly encouraged to provide more discussion on this potential contradiction and experiment the variant of ensemble in the rebuttal.
- Though the t-SNE visualizations show that the proposed method is able to produce more separation between the classes from different domains, but actually the degree of separation is not that significant (if we consider the high-level idea illustration provided in Figure.1, where the domain identification ideally would produce different subspaces across domains). The authors are highly encouraged to provide further analysis (e.g. are we able to define a metric to measure the degree of separation thus having more objective comparison between different methods) and discussion on such aspect.

---

> ### Author Response · Authors · 2022-08-02
> **Response2 to Reviewer Lxav**
>
>
> ---
> Table R3. Average accuracies (%) of the originally proposed S-liPrompts (each inference uses one single domain-selected CLIP prediction) and its ensemble version (each inference utilizes the voting strategy on all domain-based CLIP predictions) on the CDDB and DomainNet datasets for domain incremental learning.
> |           | Proposed S-liPrompts (Voting) | Proposed S-liPrompts  (Proposed Domain Selection) | Relative Improvement |
> |-----------|-------------------------------|---------------------------------------------------|----------------------|
> | CDDB      | 65.47                         | 88.65                                             | 13                   |
> | DomainNet | 58.85                         | 67.78                                             | 9                    |
> ---
>
> ### **Q3: Metrics to measure the degree of separation.**
>
>
> Thanks for raising the insightful concern. We used t-SNE as it is a popular choice for visualization. Nonetheless, as you pointed out, though the significant superiority of our proposed S-liPrompts is shown clearly via t-SNE, it is mainly in terms of domain separation rather than class separation. This phenomenon is mainly from the fact that t-SNE visualization is limited to the 2-dim projection of the original high-dimensional data. Therefore, apart from using t-SNE visualization, it is better if we can additionally use quantitative metrics to measure the degree of class separation domain by domain.
>
> Accordingly, we (re-)collected the classification/detection accuracies in Tables 1 & 2 (main paper), which reflects the average domain-wise accuracies and thus can be a favorable metric for this purpose. Following your suggestion and the related paper [32], we additionally introduce a precision based metric to measure the domain-wise class separation degree. The results in Tables R4, R5 demonstrate the consistent superiority of the proposed S-iPrompts and S-liPrompts methods in terms of the class separation using such various metrics. We included this additional study in the revised supp. material.
>
>
> ---
> Table R4. Detection accuracies (%) of the proposed S-liPrompts/S-iPrompts and the main competing methods (L2P and DyTox) on the used CDDB dataset for deepfake domain incremental learning. Task1-tas5: GauGAN, BigGAN, WildDeepfake, WhichFaceReal, SAN, which are used to train S-liPrompts on the CDDB dataset. **Bold**: best results, *Italic*: second best results.
> |             | task1 | task2 | task3 | task4 | task5 |  Average  |    Min    |    Max    |
> |-------------|:-----:|:-----:|:-----:|:-----:|:-----:|:---------:|:---------:|:---------:|
> | S-liPrompts | 99.30 | 96.75 | 82.06 | 96.25 | 68.89 | **88.65** | **68.89** | **99.30** |
> | S-iPrompts  | 90.30 | 81.88 | 72.76 | 84.25 | 43.30 |  _74.50_  |  _43.30_  |  _90.30_  |
> | L2P         | 80.73 | 62.60 | 58.98 | 57.48 | 46.59 |   61.28   |   46.59   |   80.73   |
> | DyTox       | 48.91 | 50.00 | 59.19 | 50.00 | 50.00 |   51.62   |   48.91   |   59.19   |
> ---
> ---
> Table R5. Precisions (%) of the proposed S-liPrompts/S-iPrompts and the main competing methods (L2P and DyTox) on the used CDDB dataset for deepfake domain incremental learning. Here precision is the ratio tp / (tp + fp) where tp is the number of true positives and fp the number of false positives. Task1-tas5: GauGAN, BigGAN, WildDeepfake, WhichFaceReal, SAN, which are used to train S-liPrompts on the CDDB dataset. **Bold**: best results, *Italic*: second best results.
> |             | task1 | task2 | task3 | task4 | task5 |  Average  |    Min    |    Max    |
> |-------------|:-----:|:-----:|:-----:|:-----:|:-----:|:---------:|:---------:|:---------:|
> | S-liPrompts | 99.80 | 98.45 | 81.97 | 96.54 | 75.76 | **90.50** | **75.76** | **99.80** |
> |  S-iPrompts | 90.95 | 83.64 | 73.21 | 84.08 | 43.75 |  _75.13_  |   43.75   |  _90.95_  |
> |     L2P     | 83.10 | 77.29 | 65.39 | 57.79 | 66.67 |   70.05   |  _57.79_  |   83.10   |
> |    DyTox    | 52.99 | 50.00 | 85.86 | 50.00 | 50.00 |   57.77   |   50.00   |   85.86   |
> ---

---

> ### Author Response · Authors · 2022-08-02
> **Response1 to Reviewer Lxav**
>
> We thank Reviewer Lxav for the very positive feedback and the constructive suggestions. Following the suggestions to further strengthen our paper, we discuss the questions on the potential conflict between the domain-specific prompting paradigm and the robustness to incorrect domain identification, the insignificant separation between different domains/classes as well as the improvement of using ensemble methods with more experiments.
>
>
> ### **Q1: Necessity of performing domain identification/selection.**
> ---
> Table R1. Average accuracies (%) of using random domain selection and our proposed domain selection on CDDB and DomainNet for domain incremental learning. The two domain selection cases are both based on the proposed S-liPrompts.
> |           | Random Domain Selection | Proposed Domain Selection  | Relative Improvement |
> |-----------|-------------------------|----------------------------|----------------------|
> | CDDB      | 80.12                   | 88.65                      | 8                    |
> | DomainNet | 49.94                   | 67.78                      | 18                   |
> ---
>
> ---
> Table R2: Out-of-distribution (OOD) experiment (supp. material paper Table2). Accuracies (%) of the application of the trained S-liPrompts on S1-S5 to out-of-domains OOD1-OOD3 that are 3 unseen domains used in CDDB. S1-S5: GauGAN, BigGAN, WildDeepfake, WhichFaceReal, SAN, which are used to train S-liPrompts on the CDDB dataset. OOD1-OOD3: FaceForensic++, Glow, StarGAN, which are not used to train S-liPrompts.
>
> |      | OOD1  | OOD2  | OOD3  | Avg   |
> |------|-------|-------|-------|-------|
> | CDDB | 75.69 | 64.44 | 92.46 | 77.53 |
> ---
>
> In addition to the comparison on CDDB (Table 4 in the main paper), we further compare the case of performing random domain selection against that of using our proposed domain selection on DomainNet. Table R1 reports the results on these two datasets. It shows the consistently remarkable relative improvements (by more than **8%** on CDDB, and about **18%** on DomainNet) of the proposed domain selection over the random domain selection. This considerable improvement justifies the necessity of the proposed domain selection, which attributes to the significance of the proposed independent prompting paradigm. On the other hand, the out-of-distribution (OOD) experiment in Table R2 (from supp. material Table2) verifies that the random domain selection (in Table R1) has promising results mainly due to the satisfactory ability of the proposed language-image prompt learning scheme over the CLIP model. Nonetheless, the results can be further improved significantly using the proposed domain selection (Table R1), showing the clear effectiveness of the proposed independent prompting paradigm. In the revised supp. material, we added such a study on the necessity of performing domain identification/selection.
>
> ### **Q2: Introducing ensemble methods into S-Prompts.**
>
> Following your good suggestion, we further study one of the most popular ensemble methods (i.e., voting) on the proposed S-liPrompts. Table R3 summarizes the comparison of the proposed S-liPrompts (with domain selection) against its voting-based version in terms of average accuracies on CDDB and DomainNet for domain incremental learning. The significant relative increases (by about **13%** on CDDB, **9%** on DomainNet) show that the proposed S-liPrompts method favors the proposed domain selection method that chooses the most expertised (i.e., the most related domain-based) prompting model for the inference on each test sample. The excellent performance of the selected domain-based prompting models verifies the significance of the proposed independent prompting scheme. For the voting-based inference on each test sample, we first feed all the learned domain-based prompts to the CLIP models one by one resulting in multiple predictions, and we then use the majority voting strategy on the individual results to get the final prediction. The clearly superior performance of the proposed S-liPrompts with domain selection mainly stems from the most expertised model for the given test sample. Except for this model, the rest ones are less expertised for the given test sample, and they might be dominant to corrupt the final prediction when doing the majority voting. In this case, the voting scheme could lead to performance degradation. Moreover, it requests running all the learned CLIP-based prompting models for the voting on each test sample, and thus it is much more time-consuming than the proposed inference scheme.  We added such additional experiments and discussions in the revised supp. material.

---

### Author Response · Authors · 2022-08-02
**We thank all reviewers for their positive and valuable feedback**

We are glad that the reviewers found the idea in the paper novel/simple (Reviewers Lxav and WPXn). The reviewers also find the proposed language-image prompting brand-new (Reviewers Lxav and CnBo), and the proposed method clear and elegant (Reviewer GPZj). We are also delighted that the reviewers found the experiments and analysis to be extensive (Reviewers Lxav and GPZj), and that the experimental results demonstrate the impressive/persuasive superiority (Reviewers Lxav, GPZj and CnBo) of the proposed methods compared to the state-of-the-art. The reviewers also point out that the paper is well-written and easy to follow (Reviewers Lxav, GPZj and WPXn). Below, we address all reviewers' questions and concerns.

---

### Meta-Review · Area_Chair_ksb5 · 2022-08-23

**Recommendation:** Accept
**Confidence:** Certain

**Metareview:**

This paper adopts the prompt learning idea into continual learning to tackle the problem of domain incremental learning. The proposed approach is clear, the writing is easy to follow. The experiment is convincing.

**Award:**

No

---

### Decision · Program_Chairs · 2022-09-14

Accept